# Catchments do not strictly follow Budyko curves over multiple decades but deviations are minor and predictable

Muhammad Ibrahim[1], Miriam Coenders-Gerrits[1], Ruud van der Ent[1], Markus Hrachowitz [1]

[1]Department of Water Management, Faculty of Civil Engineering and Geosciences, Delft University of Technology, Delft, The Netherlands

*Correspondence to*: Muhammad Ibrahim (m.ibrahim@tudelft.nl)

**Abstract.** Quantification of precipitation partitioning into evaporation and runoff is crucial for predicting future water availability. Within the widely used Budyko framework, which relates the long-term aridity index to the long-term evaporative index, curvilinear relationships between these indices (i.e., parametric Budyko curves) allow for the quantification of precipitation partitioning under prevailing climatic conditions. A common assumption is that movement along a specific Budyko curve with changes in the aridity index over time can be used as a predictor for catchment responses to changing climatic conditions. However, various studies have reported deviations around these curves, which raises questions about the usefulness of the method for future predictions. To investigate whether parametric Budyko curves still have predictive power, we quantified the global, regional, and local evolution of deviations of catchments from their parametric Budyko curves over multiple subsequent 20-year periods throughout the last century, based on historical long-term water balance data from over 2000 river catchments worldwide. This process resulted in up to four 20-year distributions of annual deviations from the long-term mean parametric curve for each catchment. To use these distributions of deviations to predict future deviations, the temporal stability of these four distributions of deviations was evaluated between subsequent periods of time. On average, it was found that the majority of 62 % of study catchments did not significantly deviate from their expected parametric Budyko curves. From the remaining 38 % of catchments that deviated from their expected curves, the long-term magnitude of median deviations remains minor, with 70 % of catchments falling within the range of ±0.025 of the expected evaporative index. When these median deviations were expressed as relative changes in discharge, catchments in arid regions showed higher susceptibility to larger discharge shifts compared to those in humid regions. Furthermore, a significant majority of catchments, constituting around the same percentage, were found to have stable distributions of deviations across multiple time periods, making them well-suited to statistically predict future deviations with high predictive power. These findings suggest that while trajectories of change in catchments do not strictly follow the expected long-term mean parametric Budyko curves, the deviations are minor and quantifiable. Consequently, taking into account these deviations, the parametric formulations of the Budyko framework remain a valuable tool for predicting future evaporation and runoff under changing climatic conditions, within quantifiable margins of error.

## 1 Introduction

Climate change is likely to have a profound impact on future global water resources (Jaramillo et al., 2018; Xing et al., 2018) by causing major shifts in the water balance of river basins world-wide (Serpa et al., 2015; Hattermann et al., 2017). Robust quantitative estimates of future water resources are therefore required to develop policies and to design engineering interventions that will allow the mitigation of the potentially adverse effects of these shifts on water supply (Destouni et al., 2013).

From the early 20[th] century onwards, multiple authors have suggested analytical, functionally similar non-parametric, curvilinear relationships that describe the long-term average partitioning of precipitation into runoff and evaporative fluxes in terrestrial hydrological systems (Schreiber, 1904; Oldekop, 1911; Budyko, 1948). In spite of differences in their detailed mathematical formulation (Arora, 2002; Andréassian et al., 2016), all these relationships allow to map the long-term mean fraction of precipitation $P$ that is evaporated, i.e., the evaporative index $I_E = E_A/P$, onto the long-term mean ratio of energy input, expressed as potential evaporation $E_P$, over precipitation, referred to as aridity index $I_A = E_P/P$. Many studies have demonstrated that empirical evaporative indices $I_E$ of river catchments world-wide indeed scatter rather narrowly around these non-parametric Budyko curves (Turc, 1954; Budyko, 1961; Choudhury, 1999; Zhang et al., 2001; Donohue et al., 2007; Berghuijs et al., 2014; Van Der Velde et al., 2014; Andréassian et al., 2016; Jaramillo et al., 2018; Reaver et al., 2022). To better account for the scatter, several parametric reformulations of the non-parametric Budyko curves have been proposed (Turc, 1954; Mezentsev, 1955; Tixeront, 1964; Fu, 1981). These one-parameter formulations were shown to be functionally almost equivalent to each other (Yang et al., 2008). Their parameter, hereafter referred to as $\omega$, defines catchment specific parametric Budyko curves that locate each catchment on a uniquely defined position in the space spanned by $I_A$ and $I_E$ i.e., the Budyko framework. The $\omega$ parameter is widely interpreted to encapsulate all combined properties of a catchment that may influence the storage and release of water other than $I_A$ (Milly, 1994; Donohue et al., 2012; Shao et al., 2012).

The fact that the long-term water balance exhibits such a relatively consistent behaviour across a wide spectrum of hydroclimatically and physiographically distinct environments has led to the hypothesis that the general shape of Budyko curves emerges for natural systems in a co-evolution of climate, soil water storage and vegetation properties (Milly, 1994; Porporato et al., 2004; Donohue et al., 2012; Gentine et al., 2012; Troch et al., 2013). Consequently, it may plausibly be assumed that once equilibrium is reached after a change in $I_A$, the water partitioning in a catchment converges towards a new but predictable stable state (here: $I_E$), by following its catchment specific parametric Budyko curve defined by $\omega$. By extension, such a space–time symmetry under a changing climate may then allow estimates of future $I_E$, and thus of $E_A$ and $Q$, based on changes in $I_A$, inferred from future projections of $P$ and $E_P$ (Roderick and Farquhar, 2011; Wang et al., 2016; Liu et al., 2020; Bouaziz et al., 2022).

However, parametric Budyko curves and their $\omega$ parameters were originally not developed from physical reasoning but rather from a largely process-agnostic, mathematical perspective with the aim to statistically describe observed data. They,

therefore, do not have a clearly defined physical meaning and the interaction of actual processes, that control $\omega$ in specific environments, is poorly understood. Consequently, mechanistic evidence that supports the space–time symmetry hypothesis remains erratic. This poses a serious obstacle for the formulation of a general mechanistic description to quantitatively and mechanistically link $\omega$ of parametric Budyko curves (and thus $I_E$) to catchment properties other than $I_A$ (Xu et al., 2013; Padrón et al., 2017). This further entails that estimates of $\omega$ and the associated $I_E$ for ungauged catchments or future climate conditions may be subject to major uncertainties and should therefore be interpreted from a probabilistic perspective (Greve et al., 2015).

Recently, it was also argued that catchments should not be necessarily expected to follow their long-term average, catchment specific parametric Budyko curves when subject to climatic perturbations, expressed as changes in $I_A$ (Berghuijs and Woods, 2016; Jaramillo et al., 2018; Jaramillo et al., 2022; Reaver et al., 2022). Such deviations ($\varepsilon_{IE\omega}$) from the expected parametric Budyko curve, were previously referred to as residual or landscape-driven, indicating that many factors other than $I_A$, such as human-induced changes in water and land use (e.g. afforestation, deforestation, irrigation, reservoir construction) also play a role (Donohue et al., 2007; Wang and Hejazi, 2011; Sterling et al., 2012; Destouni et al., 2013; Van Der Velde et al., 2014; Jaramillo and Destouni, 2015; Levi et al., 2015; Nijzink et al., 2016; Daly et al., 2019; Gan et al., 2021; Hrachowitz et al., 2021). Where $\varepsilon_{IE\omega}$ is defined as the absolute difference between the observed evaporative index ($I_{E,o}$) and the predicted evaporative index ($I_E$) derived from the expected parametric Budyko curve, making it dimensionless. As a consequence, Reaver et al. (2022) have warned that parametric Budyko curves may have no predictive power at all. This may be a too pessimistic perspective. First, the average magnitudes of $\varepsilon_{IE\omega}$ so far reported in studies remain rather low (e.g. Tempel et al. (2024); Wang et al. (2024)). Second, there is increasing evidence that estimates in water yield are much less sensitive to fluctuations in $\omega$ (and thus $\varepsilon_{IE\omega}$) than to changes in precipitation, in particular for humid environments (Roderick and Farquhar, 2011; Berghuijs et al., 2017). In addition, the assumption of steady conditions might not be applicable (Mianabadi et al., 2020) and the presence of uncertainties in the modelling process are inevitable (Westerberg et al., 2011; Nearing et al., 2016). In other words, some level of deviation from the parametric Budyko curves is to be expected, as different time periods will never be characterized by exactly the same environmental conditions. However, the mechanistic processes that control these deviations, and thus $\omega$, are not well understood.

Although part of several previous analyses (Destouni et al., 2013; Van Der Velde et al., 2014; Berghuijs and Woods, 2016; Jaramillo et al., 2022; Reaver et al., 2022), to our knowledge, there has been no systematic, in-depth analysis of the distributions of $\varepsilon_{IE\omega}$ or their evolution over multiple time periods at global, regional, and local scales explicitly reported in the literature. Jaramillo and Destouni (2014), Jaramillo et al. (2018) and Tempel et al. (2024) provide estimates of average $\varepsilon_{IE\omega}$ for several regions but limited their analyses to two independent time periods, while Wang et al. (2024) analysed distributions of $\varepsilon_{IE\omega}$ over multiple decades in one single river basin. In contrast, Reaver et al. (2022) quantified $\varepsilon_{IE\omega}$ over multiple time periods but explicitly reported only estimates of $\varepsilon_{IE\omega}$ maxima for individual catchments, thus describing merely the most extreme situations.

Our research question is whether the distributions and average magnitudes of $\varepsilon_{IE\omega}$ remain stable and thus probabilistically predictable over time under changing environmental conditions in space and time. A positive answer to this question would imply that parametric Budyko curves can indeed be, at least over time scales of several decades, considered useful for predicting future $I_E$ under changing conditions within quantifiable margins of error. Based on historical long-term water balance data from > 2000 river catchments worldwide, we therefore here quantify the distributions of deviations of catchments from parametric Budyko curves, i.e., $\varepsilon_{IE\omega}$, at global, regional, and local scales between multiple 20-year periods throughout the 20th century. Specifically, we test the hypothesis that the distributions of $\varepsilon_{IE\omega}$ are too wide and temporally too unstable so that $I_E$ from parametric Budyko curves needs to be considered practically unpredictable with the available data.

## 2 Datasets and methods

### 2.1 Meteorological and hydrological data

Daily precipitation $P$ [mm d$^{-1}$] as well as maximum and minimum temperature $T$ [°C] data at a spatial resolution of 0.5° x 0.5° were obtained from the Global Soil Wetness Project Phase-3 (GSWP-3); (Dirmeyer et al., 2006) and spatially averaged for each study catchment over the time period 1901 – 2015.

Potential evaporation $E_P$ [mm d$^{-1}$] was estimated based on the method proposed by Hargreaves and Samani (1982):

$$E_P = \alpha R_a (T_a + 17.8)\sqrt{(T_{\max} - T_{\min})} \qquad (1)$$

Where $\alpha \sim 0.0023$ is a constant used to convert MJ m$^{-2}$ day$^{-1}$ to mm day$^{-1}$, $R_a$ is the extraterrestrial radiation at the top of the atmosphere [MJ m$^{-2}$ day$^{-1}$] and $T_a$, $T_{\max}$ and $T_{\min}$ are the daily average, maximum and minimum temperatures [°C], respectively. $R_a$ is estimated by using the method proposed by Duffie and Beckman (1980).

In this study, we obtained annual river flow data from the Global Streamflow Indices and Metadata (GSIM) archive (Do et al., 2018; Gudmundsson et al., 2018) which consists of in situ streamflow observations data for over 30000 gauging stations worldwide. We selected stations with runoff data spanning at least 50 years in the 1901–2015 period, excluding those with a data quality flag marked as 'Caution'. After filtering, we retained 2387 river catchments with data series ranging from 50 years to 100 years (median: 78 years). These catchments vary in size, from 4 to 3,475,000 km$^2$ (median $\sim$1564 km²; Fig. 1a). The catchments represent diverse hydro-climatic conditions (Fig. 1b-f), as indicated by the long-term average aridity indices ($I_A$) that range from 0.19 to 6.66 (median: 0.97; Fig. 1e) and evaporative indices ($I_E$) that range from 0.06 to 0.99 (median: 0.65; Fig. 1f).

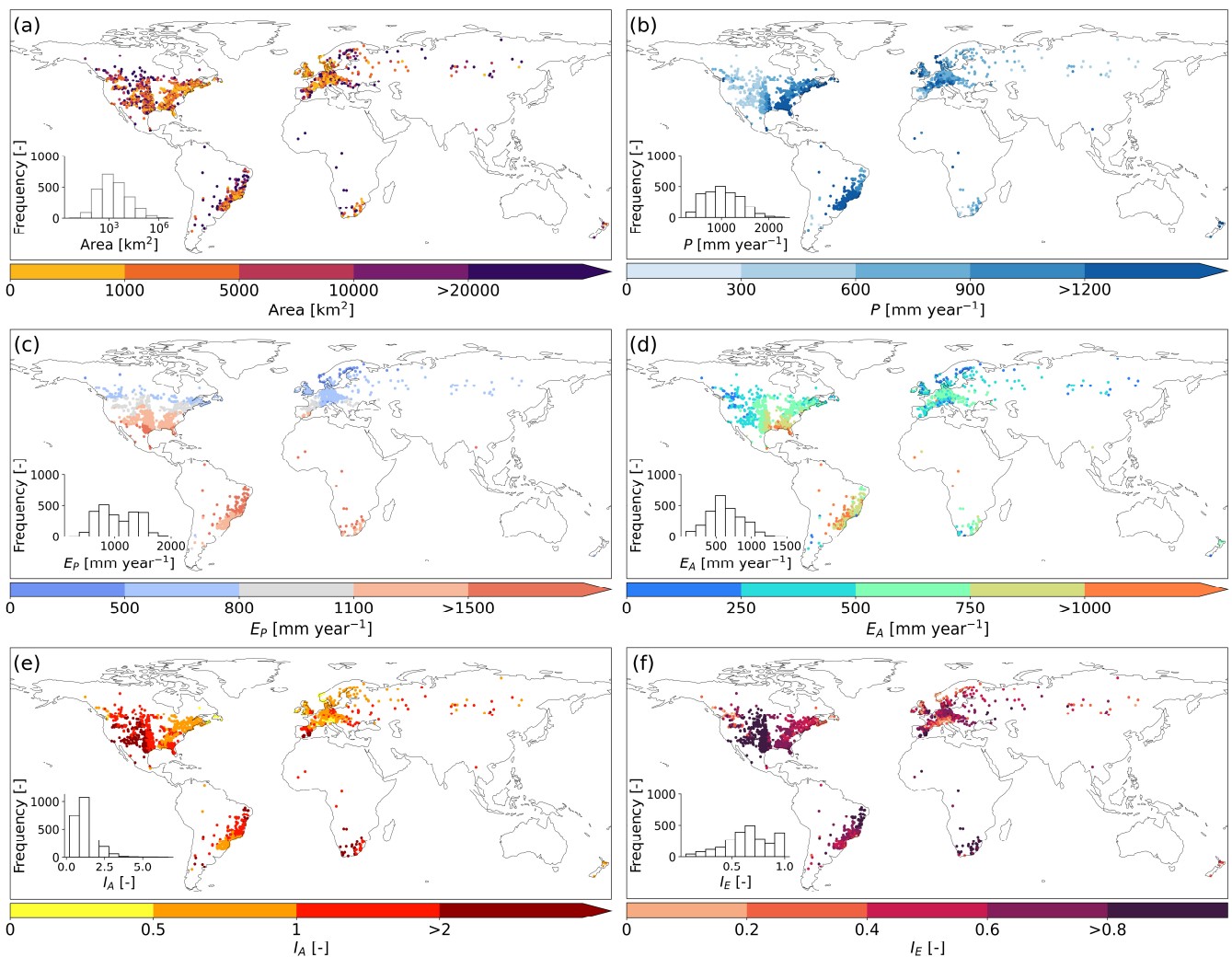

**Figure 1: Spatial distribution of 2387 studied catchments along with topographic characteristics and long-term mean (1901-2015) climatic indices: a) Catchment area, b) Precipitation *P*, c) Potential evaporation *E$_P$*, d) Actual evaporation *E$_A$* = *P-Q*, e) Aridity Index *I$_A$*, and f) Evaporative Index *I$_E$*.**

## 2.2 Methods

The subsequent experiment to estimate for each of the 2387 study catchments the deviations $\varepsilon_{IE\omega}$ from its expected evaporative indices $I_E$ over multiple subsequent time periods is based on the parametric Tixeront-Fu reformulation of the Budyko hypotheses (Tixeront, 1964; Fu, 1981), as illustrated in Fig. 2:

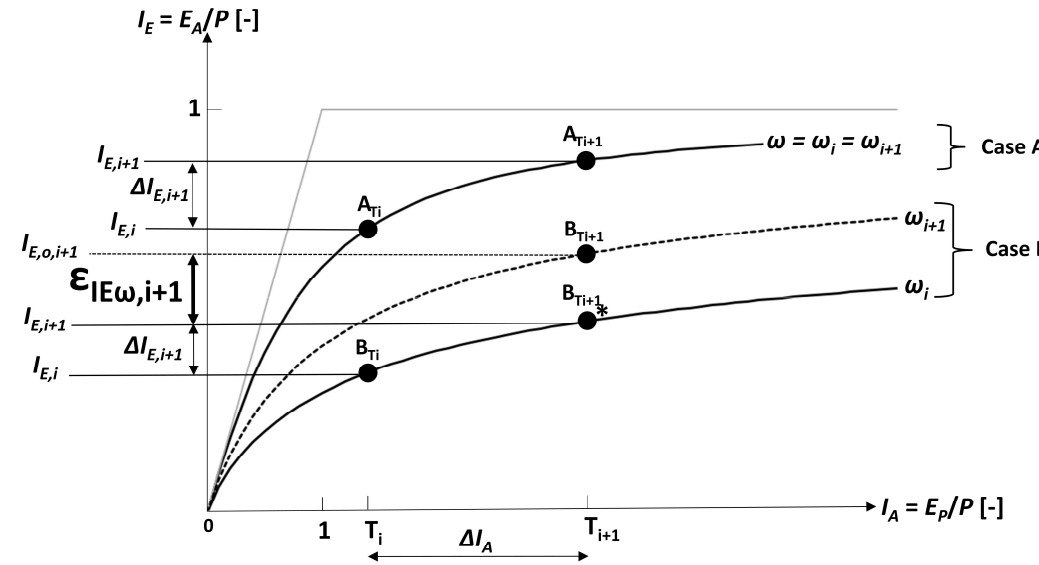

**Figure 2: A schematic representation of a catchment movement in Budyko space between two long-term time periods $T_i$ and $T_{i+1}$. Case A: Catchment A moves along the same Budyko curve from the first period $T_i$ to the next period $T_{i+1}$ (i.e., $\omega_i = \omega_{i+1}$). Case B: Catchment B has deviated from its expected parametric Budyko curve (i.e., $\omega_i \neq \omega_{i+1}$), resulting in deviation $\varepsilon_{IE\omega,i+1}$ (Eq.(4))**

This movement in Budyko space is governed by the following equation:

$$I_E = \frac{E_A}{P} = 1 + \frac{E_P}{P} - \left[1 + \left(\frac{E_P}{P}\right)^{\omega}\right]^{\frac{1}{\omega}} \tag{2}$$

where $\omega$ is a catchment-specific parameter estimated from long-term averages of observed $P$, $E_P$ and $E_A = P - Q$, assuming negligible change in storage $dS/dt$.

Equation (2) suggests that with a given $\omega$, hydro-climatic shifts between two periods $T_i$ and $T_{i+1}$, expressed as changes in aridity index $\Delta I_A = \Delta(E_P/P)$ will lead to predictable changes $\Delta I_{E,i+1}$ (Case A). In other words, catchments will follow their specific curves in period $T_{i+1}$, defined by parameter $\omega = \omega_i = \omega_{i+1}$ to an expected new $I_{E,i+1}$, which is expressed as:

$$I_{E,i+1} = I_{E,i} + \Delta I_{E,i+1} \text{ (Case A in Fig. 2)} \tag{3}$$

However, in reality, as described above, $\omega$ is often not constant over time (Case B). Catchments therefore do not strictly follow their $I_{E,i}$ curve defined by $\omega_i$ (from $T_i$) in a subsequent period $T_{i+1}$. For period $T_{i+1}$, this therefore leads to additional deviation $\varepsilon_{IE\omega,i+1}$ which is described as:

$$\varepsilon_{IE\omega,i+1} = I_{E,o,i+1} - I_{E,i+1} \neq 0 \tag{4}$$

representing the difference between the actually observed $I_{E,o,i+1}$ from the expected $I_{E,i+1}$. Thus, for period $T_{i+1}$, the observed $I_{E,o,i+1}$ depends on the combination of the predicted change and these deviations, i.e.,

$$I_{E,o,i+1} = I_{E,i} + \Delta I_{E,i+1} + \varepsilon_{IE\omega,i+1} \quad \text{(Case B in Fig. 2)} \tag{5}$$

Here, we have sub-divided the available data records of each catchment into up to five individual 20-year periods over the last century, denoted as $T_i$ (Table 1), where $T_i$ represents the $i^{th}$ 20-year period. This 20-year period was chosen deliberately to balance the need for a sufficiently long period to minimize the impact of storage changes, while preserving the temporal sequence in the data that allowed us to place each catchment into a specific temporal stability category (as described in *Step-4*). We assume that 20-year periods are long enough to satisfy d$S$/d$t \approx 0$, supported by Han et al. (2020), who demonstrated that in more than 80 % of catchments worldwide, d$S$/d$t$ is less than 5 % over 20-year periods. Using longer periods, such as 30 years as used in previous studies (e.g. Destouni et al. (2013)), would have smoothed out potential shifts and limited the ability to detect systematic changes. In addition, 20-year periods align with planning horizons in many water resource management decisions.

**Table 1: Symbols used in this study to present 20-year periods ($T_i$), changes between subsequent 20-year periods and distributions of deviations.**

| Time period | Symbols | | |
|---|---|---|---|
| | 20-year periods | Change between subsequent 20-year periods | Distributions of deviations |
| 1901-1920 | $T_1$ | $\Delta_{1-2}$ | $\varepsilon_{IE\Delta 1}$ |
| 1921-1940 | $T_2$ | | |
| | | $\Delta_{2-3}$ | $\varepsilon_{IE\Delta 2}$ |
| 1941-1960 | $T_3$ | | |
| | | $\Delta_{3-4}$ | $\varepsilon_{IE\Delta 3}$ |
| 1961-1980 | $T_4$ | | |
| | | $\Delta_{4-5}$ | $\varepsilon_{IE\Delta 4}$ |
| 1981-2000 | $T_5$ | | |

The experiment to estimate deviations $\varepsilon_{IE\omega}$ between the five individual periods $T_1$–$T_5$ for the study catchments was then carried out in a systematic sequence of 5 specific steps as illustrated in Fig. 3 and described in the following:

*Step 1: Estimation of catchment-specific $I_{E,i}$ curves and the distribution of annual $I_{E,o}$ around it for each period* $T_i$

For each catchment and each individual 20-year time period $T_i$, the catchment-specific parametric Budyko curve $I_{E,i}$ defined by parameter $\omega_i$ is obtained by fitting Eq.(2) to the set of 20 annual values of each catchment in the Budyko space, as computed from the observed water balance data. The decision to obtain the $\omega_i$ for each 20-year period by fitting Eq.(2) to the

set of $n = 20$ corresponding observed annual $I_{E,o}$ values instead of directly to their 20-year averages was a deliberate choice.

The fluctuations of the $n = 20$ annual $I_{E,o}$ values explicitly represent annual storage changes $dS/dt$ between individual years. This subsequently allowed to treat the observed annual $I_{E,o}$ probabilistically as distributions around their expected values for that period $T_i$ as defined by $I_{E,i}$ curve (Fig. 3a).

*Step 2: Distributions of annual deviations $\varepsilon_{IE\Delta j}$ from expected $I_{E,i+1}$ between subsequent time periods*

For each catchment we then used $\omega_i$ from each time period $T_i$ to compute the expected $I_{E,i+1}$ for the subsequent period $T_{i+1}$ (i.e. point $B_{Ti+1}$* in Fig. 2). This then allowed to estimate the individual deviations of the 20 annual observed $I_{E,o}$ values from the expected $I_{E,i+1}$ curve. For each pair of time periods $T_i$–$T_{i+1}$ (i.e. $T_1$–$T_2$, $T_2$–$T_3$, etc., hereafter referred to as $\Delta_{1-2}$, $\Delta_{2-3}$, etc.) this resulted in an individual distribution of annual deviations $\varepsilon_{IE\Delta j}$ around a 20-year average in each catchment (Fig. 3b). This approach using a temporally changing (dynamic) baseline was chosen as it is more sensitive to capture trends and shifts

in hydrological behaviour of catchments over time than a fixed baseline. For completeness, we also performed the same analysis by using a fixed baseline (i.e., using the earliest available period as a fixed baseline) and provide the results thereof in the Supplement.

    Note, that catchments with data for all five time periods $T_1$–$T_5$, have the maximum of $j = 4$ distributions $\varepsilon_{IE\Delta j}$. In contrast, catchments with data for only two periods, e.g., $T_2$ and $T_3$, feature only $j = 1$ distribution of between-period deviations

$\varepsilon_{IE\Delta j}$.

    Non-parametric Wilcoxon Signed Rank Tests were then used to test for each distribution $\varepsilon_{IE\Delta j}$ the null hypothesis that the median deviation is not significantly different from zero. The lower the p-value, the higher the probability that the median deviation of $\varepsilon_{IE\Delta j}$ of observed $I_{E,o,i+1}$ from expected $I_{E,i+1}$ is higher than zero for the comparison of $\varepsilon_{IE\Delta j}$ between periods $T_i$ and $T_{i+1}$.


*Step 3: Fit parametric distributions to the empirical distributions of annual deviations $\varepsilon_{IE\Delta j}$*

    For each catchment we have then fitted Skew Normal Distributions to each of the $j = 1$–4 empirical distributions of deviations $\varepsilon_{IE\Delta j}$ (Fig. 3c). The probability density function (PDF) of the skew normal distribution is given by:

$$f(x) = \frac{2}{\lambda} \phi \left( \frac{x-\xi}{\lambda} \right) \Phi(\alpha) \left( \frac{x-\xi}{\lambda} \right) \tag{6}$$

Where $\phi$ is the standard normal PDF, $\Phi$ is the standard normal cumulative distribution function, $\lambda$ is a scale parameter, $\xi$ location parameter and $\alpha$ is a shape parameter.

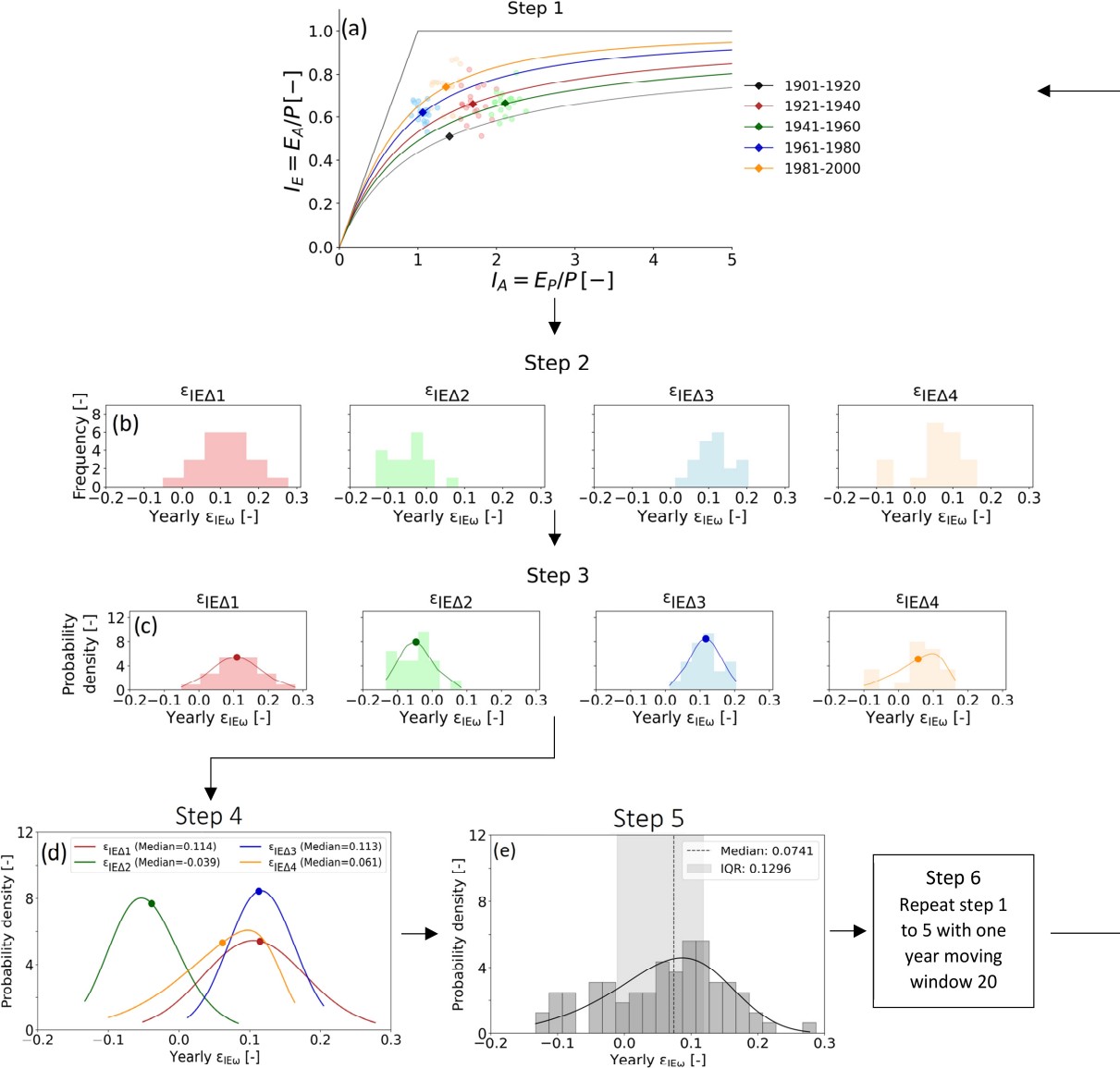

**Figure 3: Flow chart of methodology. Step 1: Estimation of catchment-specific $I_{E,i}$ curves and the distribution of annual $I_{E,o}$ around it for each period $T_i$. Step 2: Distributions of annual deviations $\varepsilon_{IE\Delta j}$ from expected $I_{E,i+1}$ between subsequent time periods. Step 3: Fit parametric distributions to the empirical distributions of annual deviations $\varepsilon_{IE\Delta j}$. Step 4: Evaluate temporal stability of the distributions $\varepsilon_{IE\Delta j}$ in subsequent pairs of time periods. Step 5: Aggregated long-term marginal distribution of annual deviations $\varepsilon_{IE\omega}$ from expected $I_E$ for each catchment. Step 6: Evaluation of the sensitivity of the marginal distributions of annual deviations $\varepsilon_{IE\omega}$ to the choice of 20-year averaging window. Note, the generated distributions of $\varepsilon_{IE\omega}$ are illustrative examples that are not based on real data.**

The mean of the distribution is computed as:

$$\overline{f(x)} = \xi + \left(\sqrt{2/\pi}\right)\frac{\lambda\alpha}{\sqrt{1+\alpha^2}} \tag{7}$$

and the standard deviation is represented by:

$$\sigma = \lambda\sqrt{\left(1 - \frac{2\alpha^2}{(1+\alpha^2)\pi}\right)} \tag{8}$$


*Step 4: Evaluate temporal stability of the distributions $\varepsilon_{IE\Delta j}$ in subsequent pairs of time periods*

For distributions of past deviations to be used to estimate deviations $\varepsilon_{IE\omega}$ under projected hydro-climatic future conditions, it is necessary to upfront evaluate whether it is plausible to assume that they retain sufficient explanatory power

under future conditions or if there is evidence against that. This was here done by analysing how stable the individual distributions in a catchment are over time. To do so, for each catchment the up to j = 4 distributions of deviations $\varepsilon_{IE\Delta j}$ from expected $I_{E,i+1}$ between subsequent time periods were compared and analysed for their changes over time (Fig. 3, Sub-steps 4.1-4.3). We have followed three sub-steps:

Sub-step 4.1

At first, non-parametric Kolmogorov-Smirnov Tests with a significance level of 5 % were used on consecutive pairs of distributions, i.e. $\varepsilon_{IE\Delta j}$ and $\varepsilon_{IE\Delta j+1}$ to test the null hypothesis that the distributions are not significantly different from each other. In case the null hypothesis is not rejected (p > 0.05), hereafter referred to with symbol "o", we consider the catchment is stable over time and in such a case the past distributions may be directly used to estimate $\varepsilon_{IE\omega}$ under future conditions with

some confidence.

Sub step 4.2

If for a catchment significant differences between consecutive pairs of distributions (p ≤ 0.05) were found, it was further analysed whether the differences can be considered arbitrarily variable or whether there is indicative evidence for the potential presence of fluctuations or systematic shifts over time. Thus, in a second step, we have checked if the median of $\varepsilon_{IE\Delta j}$

systematically decreased ("–") or increased ("+") over time. If the difference between three or more of the j distribution medians were characterized by the same sign, i.e., "–" or "+", this may be evidence for a systematic and thus non-variable shift in the median of $\varepsilon_{IE\Delta j}$ over time. In that case, past distributions $\varepsilon_{IE\Delta j}$ need to be assumed to have limited predictive power for estimating future $\varepsilon_{IE\omega}$.


Sub step 4.3

In the alternative case, when less than three distributions showed the same sign, we have in a third step analysed, whether $\varepsilon_{IE\Delta j}$ for $\Delta_j$ is influenced by the magnitude of $I_{E,i}$ and that e.g., after a 20-year period with a low $I_{E,i}$, further future

decreases and thus negative $\varepsilon_{IE\Delta j}$ are unlikely and $I_{E,i+1}$ will, more probably, swing back to higher values and thus positive $\varepsilon_{IE\Delta j}$. Similar to above, if the median $\varepsilon_{IE\omega}$ systematically decreased ("–") or increased ("+") with $I_{E,i}$ for three or more of the pairs of time periods $j$, this may be evidence for a systematic and thus non-variable shift in the median $\varepsilon_{IE\omega}$ over time, indicating limited predictive power.

**Table 2: Decision criteria to classify the time stability of the j distributions $\varepsilon_{IE\Delta j}$ for each catchment into one of the four qualitative categories "Stable", "Variable", "Alternating", "Shift" and the associated predictive power of the marginal distribution of $\varepsilon_{IE\omega}$ of a catchment, aggregating all j distributions of that catchment.**

| Tag | Kolmogorov-Smirnov Test | Systematic shift of median | Relation between $I_{E,i}$ and $\varepsilon_{IE\omega}$ | Examples | Predictive power | No. of catchments |
|---|---|---|---|---|---|---|
| Stable | $p > 0.05$ | No | No | "o o o o" or "– o o o" | High | 1651 |
| Variable | $p \leq 0.05$ | No | No | "+ – + –" or "– + + –" | Moderate | 455 |
| Alternating | $p \leq 0.05$ | No | Yes | "+ – + –" or "– + + –" | Low | 179 |
| Shift | $p \leq 0.05$ | yes | - | "– – – –" or "– + + +" | Low | 102 |

Following the above, each catchment was classified into one of four qualitative categories of temporal stability of

$\varepsilon_{IE\omega}$ (Table 2). Note, that the use of formal quantitative statistical test was here hindered by the small sample size of a maximum of four pairs of time periods and thus omitted. The temporal stability was ranked as "Stable", if between more than half of the $j$ distributions in a catchment no significant differences in median $\varepsilon_{IE\omega}$ was found, e.g., "o o o o" or "– o o o". A catchment was ranked as "Variable", if it showed an alternating sequence and thus *no* systematic shift of median $\varepsilon_{IE\omega}$ over time e.g., "+ – + –" or "– + + –" *and no* relation between $I_{E,i}$ and median $\varepsilon_{IE\omega}$. In contrast, if a catchment was characterized by an alternating

sequence and a dependency between $I_{E,i}$ and median $\varepsilon_{IE\omega}$, it was tagged as "Alternating". If, finally, between three or more of the j consecutive distributions in a catchment the median $\varepsilon_{IE\omega}$ was found to increase or decrease, e.g., "– + + +" or "– – – –",

this may indicate the presence of a systematic shift over time and the temporal stability of deviations from expected $I_E$ was tagged as "Shift".

*Step 5: Aggregated long-term marginal distribution of annual deviations $\varepsilon_{IE\omega}$ from expected $I_E$ for each catchment*

In this step the up to $j = 4$ distributions $\varepsilon_{IE\Delta j}$ were aggregated into one marginal distribution of $\varepsilon_{IE\omega}$ for each catchment (Fig. 3e). This distribution reflects the historical range of fluctuations in annual $\varepsilon_{IE\omega}$ based on all available information for each catchment. Consequently, the median $\varepsilon_{IE\omega}$ of the distribution in each catchment represents a measure of uncertainty around expected future $I_E$ based on current estimates of $\omega$ for each catchment, thereby making $I_E$ statistically predictable.

To account for the potential effect of systematic shifts in distributions $\varepsilon_{IE\Delta j}$ (Step 4) on the predictive power of the associated marginal distribution of deviations $\varepsilon_{IE\omega}$, we have tagged the marginal distribution of each catchment with a qualitative robustness flag as defined in Step 4. "Stable" distributions are characterized by the highest predictive power, distributions with "Variable" fluctuations can be expected to have moderate predictive power, while distributions tagged as "Alternating" or "Shift" do in the absence of more detailed data have rather low predictive power (Table 2).


*Step 6: Evaluate the sensitivity of the marginal distributions of annual deviations $\varepsilon_{IE\omega}$ to the choice of 20-year averaging window*

To further quantify the sensitivity of the above aggregated, i.e. marginal distributions to the choice of the individual 20-year averaging time periods, we have, in a last step, repeated the above Steps 1–5 twenty times to test all possible sequences

of 20-year periods. More specifically, in a moving window analysis we first shifted each time period $T_1$–$T_5$ by one year, i.e. 1902–1921 ($T_1$), 1922–1941 ($T_2$), etc. and repeated the above Steps 1–5. Subsequently we shifted $T_1$–$T_5$ by another year to 1903–1922 ($T_1$), 1923–1942 ($T_2$), etc. and again repeated Steps 1–5. This was done twenty times until all years of the first period, i.e. 1901–1920, were the starting years of $T_2$.

## 3 Results

### 3.1 Changes in hydro-climatic variables and movement in Budyko space (Step 1)

Throughout the 100-year study period and across all study catchments, considerable hydro-climatic variability was observed, with some variables exhibiting trend-like behaviour over time and others more cyclic behaviour. Overall, mean annual precipitation over the individual 20-year periods systematically increased by $\sim 18.4$ mm century$^{-1}$, on average (Fig. 4a), with 57 % of the catchments showing an increase between $T_1$ and $T_2$ ($\Delta_{1\text{-}2}$) and 83 % for $\Delta_{4\text{-}5}$. In contrast, mean annual

temperatures (Fig. 4b) and the associated potential evaporation (Fig. 4c) were characterized by a more fluctuating pattern. These combined factors led to slightly more arid conditions in the first half of the 20[th] century, followed by a considerable

reduction of aridity index $I_A$ and thus to a shift towards somewhat more humid conditions towards the end of the century across all of the temporal stability categories (Fig. 4e), in which, on average 78 % and 75 % of the catchments showed decreases in $I_A$ for $\Delta_{3\text{-}4}$ and $\Delta_{4\text{-}5}$, respectively. The changes in $I_A$ were accompanied by related changes in potential evaporation $E_P$ and precipitation (Fig. 4c,a). The overall movement of catchments in the Budyko space due to hydro-climatic changes are illustrated in Fig. S1 (Jaramillo and Destouni, 2014). If these movements were driven only by changes in $I_A$, catchments would be expected to move within the directional range of $45° < \alpha < 90°$ or $225 < \alpha < 270°$ (Jaramillo et al., 2022). However, observed movement of catchments are also found in other directions, indicating deviations ($\varepsilon_{IE\omega} \neq 0$) from the expected $I_E$, as elaborated in detail in Fig. S1

It is worth mentioning here that the sample sizes vary between individual 20-year periods of comparison due to the length of the data availability. Therefore, to distinguish whether the climatic variability in Fig. 4 is associated to the hydroclimatic variables or due to the change in sample size, the same plot for the catchments that are present in all periods of comparison ($n = 142$) is provided in Fig. S2. It is found that the overall pattern of temporal variability in that sub-sample largely reflects that of the full sample.

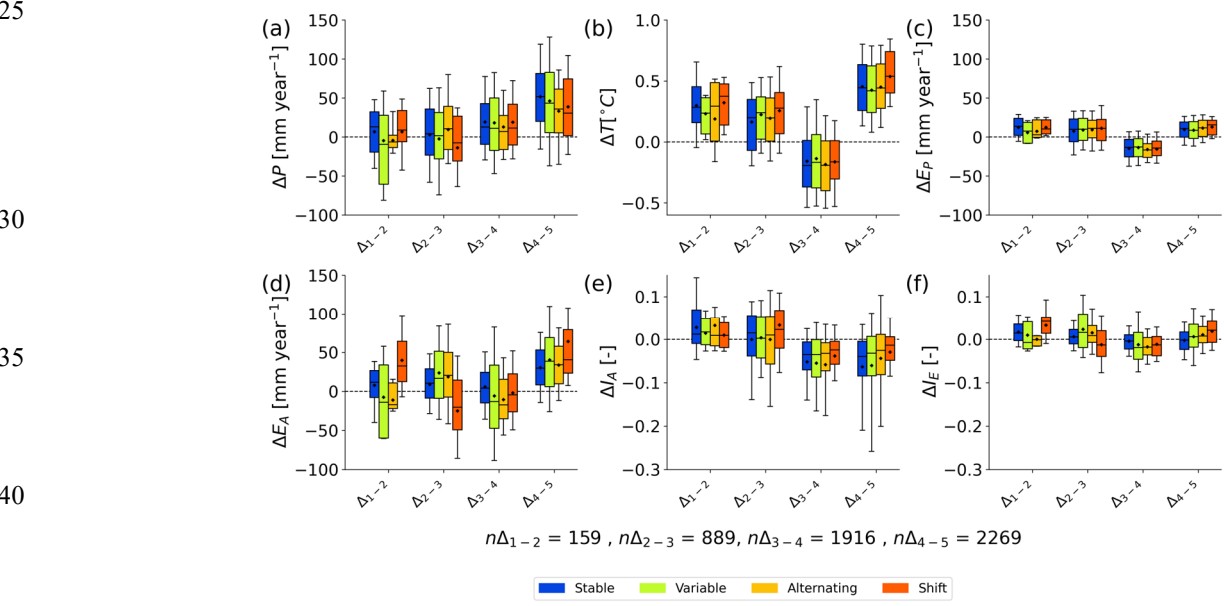

$n\Delta_{1-2} = 159$ , $n\Delta_{2-3} = 889$, $n\Delta_{3-4} = 1916$ , $n\Delta_{4-5} = 2269$

Stable | Variable | Alternating | Shift

**Figure 4: Temporal stability category-wise mean 20-year changes in hydro-climatic variables for the studied catchments between two consecutive periods. a) Precipitation $P$, b) Temperature $T$, c) Potential evaporation $E_P$, d) Actual evaporation $E_A$, e) Aridity index $I_A$, and f) Evaporative index $I_E$. The boxes represent the 25th to 75th percentiles, while whiskers extend to the 10th and 90th percentiles. Diamonds denote the arithmetic mean, and outliers are not shown.**

**3.2 Distributions of annual deviations $\varepsilon_{IE\Delta j}$ from parametric Budyko curve (Steps 2 & 3)**

The indicative evidence for presence of deviations $\varepsilon_{IE\omega} \neq 0$ in at least some catchments is further supported by a more detailed analysis of the distributions of annual deviations $\varepsilon_{IE\Delta j}$ between the pairs of subsequent 20-year periods for each of the 2387 study catchments. The results of the Wilcoxon Signed Rank Tests indeed suggest that it is likely that the median $\varepsilon_{IE\omega} \neq 0$ for a significant proportion of catchments. For example, at a 95 % confidence level (i.e., $p \leq 0.05$), 34–42 % of the distributions can be considered to feature deviations with a median $\varepsilon_{IE\omega} \neq 0$ (Fig. 5a). Conversely, this also entails that for a majority of 58–66 % of the distributions there is less evidence (i.e., $p > 0.05$) that the median $\varepsilon_{IE\omega}$ are different form zero. Note that minor $\varepsilon_{IE\omega}$ were observed in most catchments. Although these $\varepsilon_{IE\omega}$ were not classified as significant based on the Wilcoxon Signed Rank Test used here, it may be too naive to assume that the deviations $\varepsilon_{IE\omega}$ are strictly zero, as also demonstrated by Reaver et al. (2022). Overall, this is consistent with results from previous studies and shapes a picture in which catchments do not strictly and necessarily follow their expected parametric $I_E$ curves, but that the deviations thereof remain close to zero or very limited for many catchments.

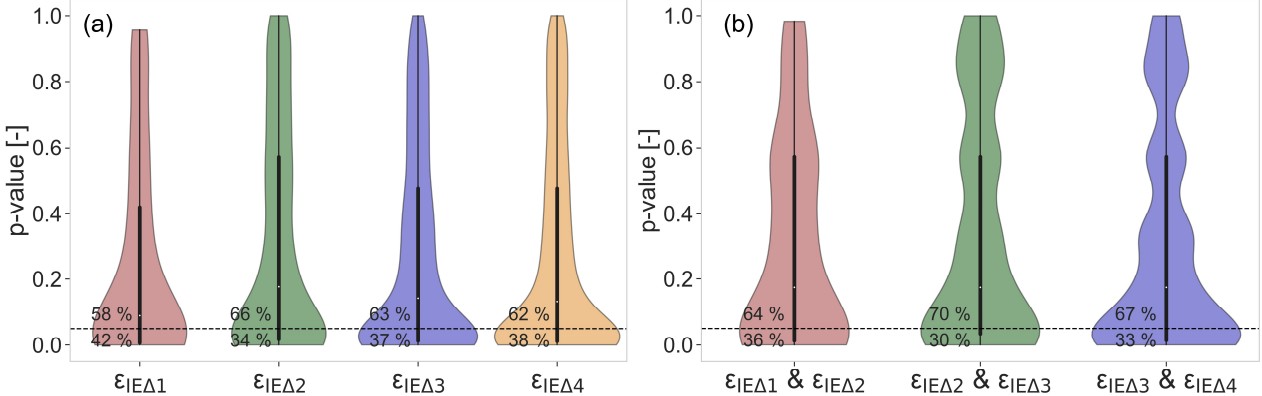

**Figure 5: Distribution of p-values from statistical tests a) Wilcoxon Signed Rank Test performed to test whether the long-term median $\varepsilon_{IE\omega}$ of the individual 20-year distributions $\varepsilon_{IE\Delta j}$ is significantly different from zero. The percentage above the significance line (0.05) shows data points where median $\varepsilon_{IE\omega}$ are not significantly different from zero, while the percentage below indicates those that are significantly different b)Kolmogorov-Smirnov Test performed on the distributions of two consecutive time periods ($\varepsilon_{IE\Delta j}$ and $\varepsilon_{IE\Delta j+1}$) to test whether the two distributions of deviations are significantly different from each other. Here, the percentage above the significance line presents data points where the distributions of two consecutive time periods are not significantly different from each other, while the percentage below indicates significant differences. Each violin plot displays the distribution of p-values, with the central black box representing the interquartile range (25th to 75th percentiles) and whiskers extending to the smallest and largest data points within 1.5 times the interquartile range. The dotted black line represents the significance level of 0.05.**

A characteristic selected example for the latter is the sequence of the four distributions of annual deviations in the Chemung River at Chemung (New York; 6455 km$^2$; ID US_0000832) across the four pairs of subsequent 20-year periods over the 20th century (Fig. 6a-c). The annual deviations $\varepsilon_{IE\omega}$ with an interquartile range of IQR $\sim$ 0.062 (Fig. 6c) concentrate quite narrowly around the medians. The medians themselves range between merely $\varepsilon_{IE\omega}$ = -0.006–0.012 (Fig. 6b). The associated p-

values from the Wilcoxon Signed Rank Test (p= 0.27–0.84) further suggest that there is only limited evidence that the median deviations $\varepsilon_{IE\omega}$ of distributions $\varepsilon_{IE\Delta j}$ are different from zero. In spite of a somewhat wider spread with an IQR ~ 0.094 (Fig. 6f), a similar pattern with consistently low median $\varepsilon_{IE\omega}$= -0.017–0.018 (Fig. 6e) (p = 0.09–0.47) was observed in the second selected example, the Lee River (Ireland; 1019 km$^2$; ID GB_0000078).

In contrast, more variable patterns were found for other catchments (Fig. 6g-o). For example, in the Sava River at Radece (Slovenia; 6004 km$^2$; ID SI_0000007) the all four distributions of the annual deviations display a wider spread, with IQR ~ 0.113, indicating a higher degree of storage fluctuation between individual years (Fig. 6i). This variability may largely be attributed to hydropower developments and the associated changes in hydropower production levels (Levi et al., 2015), which disrupt natural flow regimes by increasing runoff during high demand and altering seasonal flow patterns (Renofalt et

al., 2010; Lee et al., 2023). In addition, the medians do considerably deviate from zero, as indicated by median $\varepsilon_{IE\omega}$ ranging between -0.023 and 0.118 (Fig. 6h).

The set of 20-year average $I_E$ values and the associated parameters of the fitted parametric distributions of deviations for each of the time periods in the individual study catchments are provided in the Supplementary data downloadable from Zenodo (https://doi.org/10.5281/zenodo.14060926).

**3.3 Temporal stability of the distributions $\varepsilon_{IE\Delta j}$ (Step 4)**

     Based on the Kolmogorov-Smirnov Tests, it was found that overall 68 % of the distributions $\varepsilon_{IE\Delta j}$ between consecutive pairs of time periods are not significantly different from each other (p > 0.05; Fig. 5b). Following the criteria defined in Sect. 2.2, this resulted in 1651 catchments classified as "Stable" (Table 2; Fig. 7a). For these catchments, together corresponding to ~ 70 % of all 2387 study catchments, their respective marginal distribution of $\varepsilon_{IE\omega}$ can thus be plausibly considered to have

395 rather high predictive power. Example cases are the Chemung and Lee Rivers (Fig. 6a-f), which are characterized by sequences "o o o o" and "o o o o", respectively.

     Similarly, 455 additional catchment (19 % of all study catchments), whose distributions exhibited fluctuations over time (Kolmogorov-Smirnov Test p ≤ 0.05), but which featured only limited evidence for both, the presence of systematic shifts over time as well as for a dependency between $I_{E,i}$ and $\varepsilon_{IE\omega}$, were tagged as "Variable". An example of such a case is shown in

Fig. 6g-i for the Sava River at Radece (Slovenia; 6004 km$^2$; ID SI_0000007). It can be seen that the distributions $\varepsilon_{IE\Delta j}$ between the four pairs of time periods vary considerable with medians ranging from -0.023 and 0.118. However, the fluctuations appear to occur alternatively (" - + - +") and do suggest neither the presence of a systematic shift over time (Fig. S4b) nor a dependency between $I_{E,i}$ and median $\varepsilon_{IE\omega}$ (Fig. S4a). Despite these fluctuations, the marginal distribution of $\varepsilon_{IE\omega}$ aggregated from the 4 individual distributions $\varepsilon_{IE\Delta j}$ (Fig. 6i) can, although wider than for catchments tagged as "Stable", thus be assumed to be an at

least moderately robust representation of $\varepsilon_{IE\omega}$.

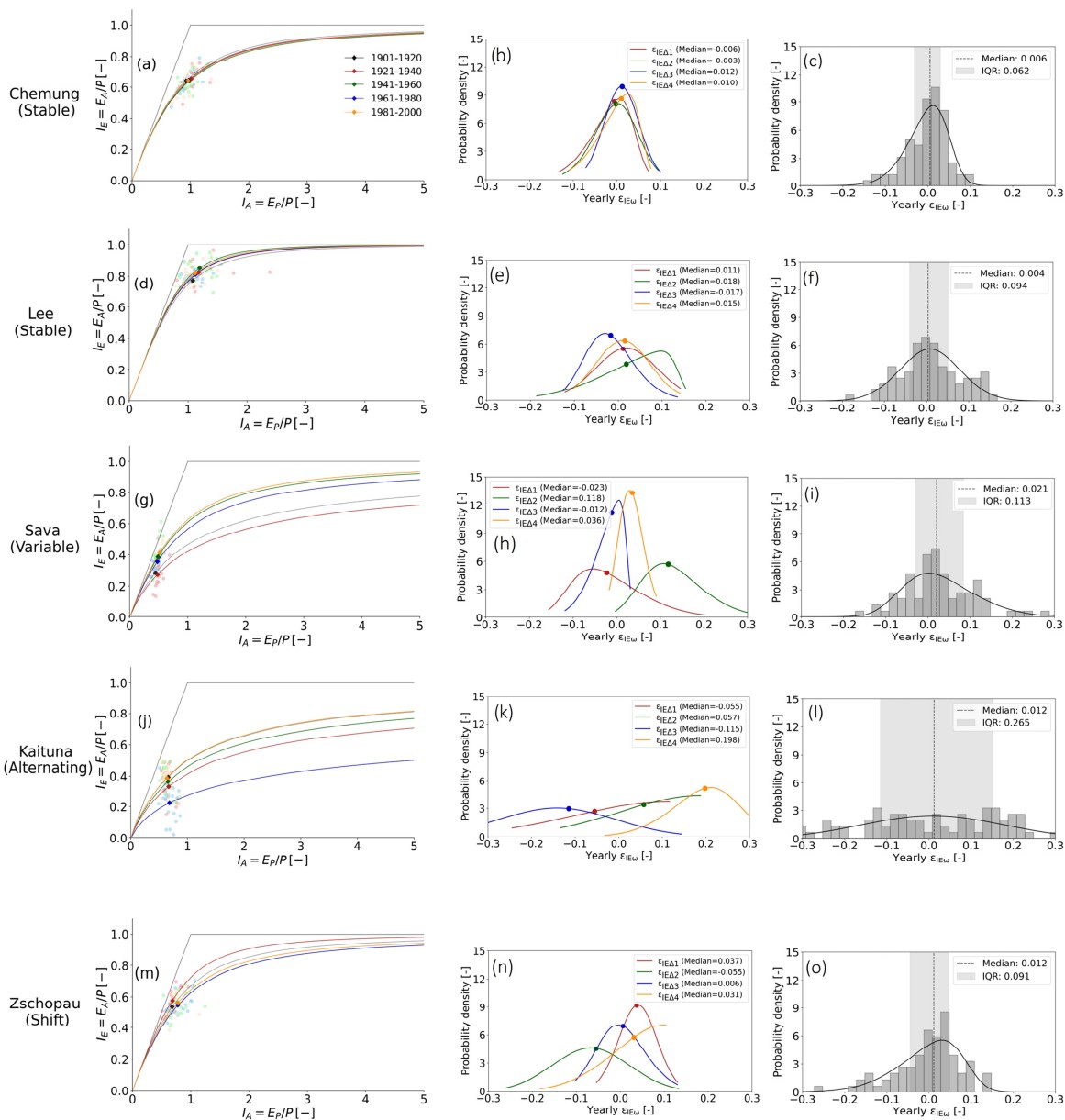

**Figure 6: Mean annual position of catchments (light colour dots) in Budyko space along with long-term mean (dark colour dots) and expected parametric Budyko curves (left column). Individual distribution of deviations (ε$_{IEΔ1}$, ε$_{IEΔ2}$, ε$_{IEΔ3}$ and ε$_{IEΔ4}$) with long term median deviation ε$_{IEω}$ values (middle column) and long-term marginal distribution of annual deviations along with long-term median values of ε$_{IEω}$ and IQR of ε$_{IEω}$ values (right column) for five example catchment: Chemung River (a-c), Lee River (d-f), Sava River (g-i), Kaituna River (j-l), and Zschopau River (m-o).**

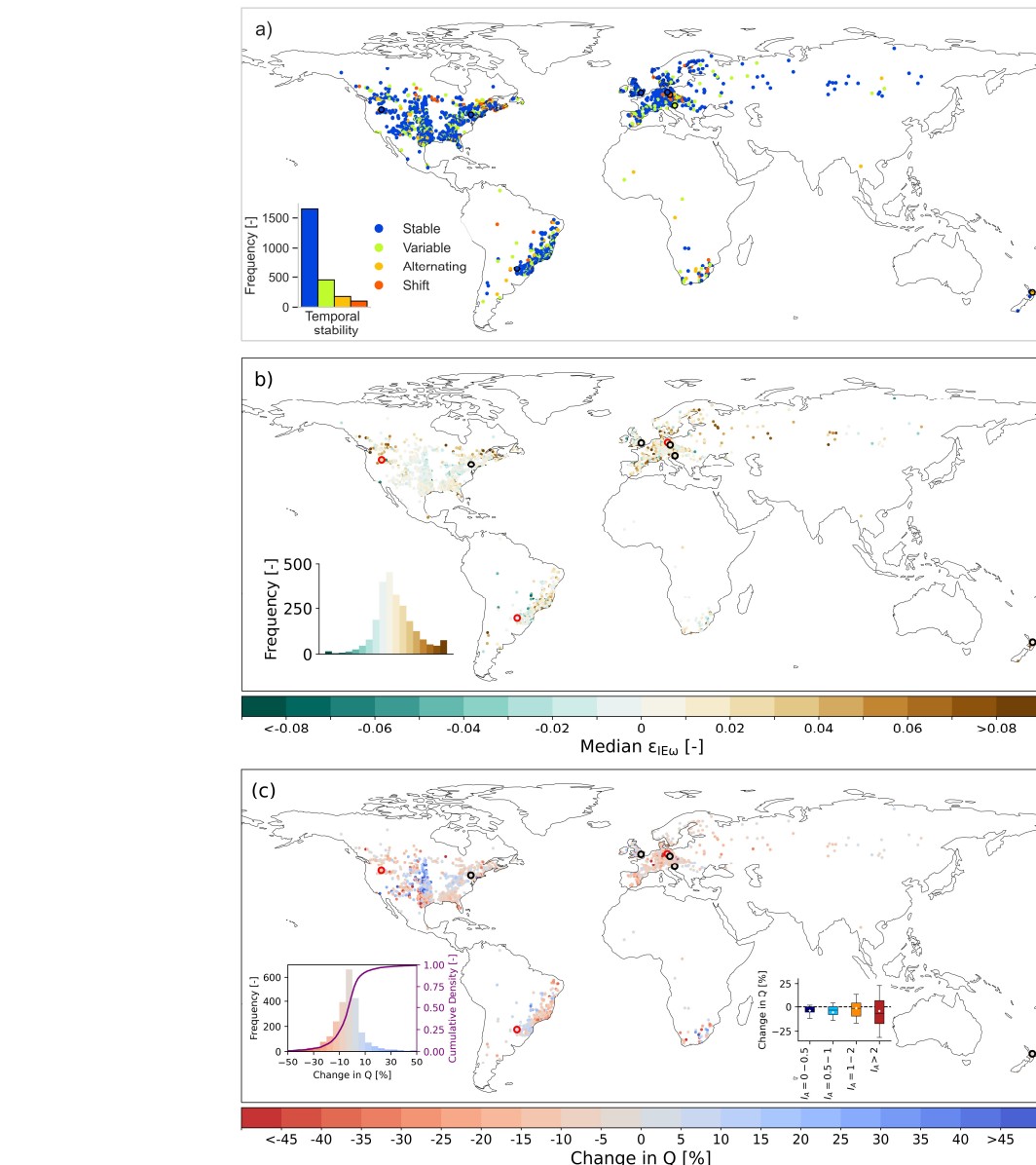

**Figure 7: a) Temporal stability, b) long-term median $\epsilon_{IE\omega}$ values map of aggregated long-term marginal distributions for the study catchments, and c) change in Q as a result of long-term median $\epsilon_{IE\omega}$ values. Histogram and cumulative density of change in Q, and change in Q across different $I_A$ bins are presented as two insets. Change in Q reflects the change due to median deviations $\epsilon_{IE\omega}$ from the expected parametric Budyko curve only (i.e., excluding any change resulting from a change in aridity and its associated movement along the expected curve). Catchments highlighted with a black border represent the 5 selected examples from Fig. 6, while those outlined in red denote three additional selected example catchments shown in the Supplement (Fig. S5). The boxes represent the 25th to 75th percentiles, while whiskers extend to the 10th and 90th percentiles. Diamonds denote the arithmetic mean, and outliers are not shown.**

In contrast, 7 % of the catchments were tagged as "Alternating" and a dependency between $I_{E,i}$ and $\varepsilon_{IE\omega}$ could not be ruled out. A characteristic example for this type of catchments is the Kaituna catchment (New Zealand; 706 km$^2$, ID NZ_0000003) in Fig. 6j-l. This catchment features major fluctuations with median $\varepsilon_{IE\omega}$ between -0.115 and 0.198. In addition, although no systematic evolution of median $\varepsilon_{IE\omega}$ over time was evident (Fig. S4d), the data suggest the potential presence of a dependency on $I_{E,i}$ as shown in Fig. S4c. The pronounced alternating behaviour of the $\varepsilon_{IE\omega}$ fluctuations between -0.115 and 0.198, could not be readily explained by factors such as land use changes as estimated from the Hilda+ gridded land cover product (Winkler et al., 2021), Seasonality Index (SI) of liquid precipitation input (i.e., rainfall + snowmelt), Parde Coefficients or median rainfall intensity (Fig. S3a,c,e-f). The SI was calculated using the formula proposed by Gao et al. (2014). A higher SI value indicates that most of the precipitation falls within a few months, while a lower value reflects more evenly distributed precipitation throughout of the year. This suggests that other additional drivers, or a combination of drivers, influence this catchment's alternating behaviour.

The remaining 102 catchments (4 %) were tagged as "Shift", as they exhibit a rather consistent shift of median $\varepsilon_{IE\omega}$ over time. This can be seen for a selected example in Fig. 6m-o. The median $\varepsilon_{IE\omega}$ in this catchment of the Zschopau River (Germany; 1544 km2; ID DE_0000027) does not only significantly vary between -0.055 and 0.037 but it does so rather systematically into one dominant direction after $\varepsilon_{IE\Delta1}$ ("+ - ++"; Fig. 6n). This shift aligns with a gradual decrease in the 20-year Seasonality Index (SI) (Fig. S3e) of liquid precipitation input (i.e., rainfall + snowmelt). In the Zschopau river catchment, this decrease in SI towards the end of the century signifies a shift towards a more evenly distributed precipitation pattern. These changes coincide with an increase in forest cover towards the end of century, as estimated from Hilda+ data (Fig. S3b). Additionally, Renner et al. (2014) and Renner and Hauffe (2024) reported a gradual recovery of forests in the Zschopau catchment during this period, which may further contribute to the observed shift.

As can be seen in Fig. 7a, the time stability of the study catchments is geographically rather homogenously distributed. Catchments tagged as "Stable" and "Variable" can be found globally, while also no regional concentrations of catchments tagged as "Alternating" and "Shift" could be identified.

### 3.4 Aggregated long-term marginal distribution of annual deviations $\varepsilon_{IE\omega}$ (Step 5)

By aggregating its j individual distributions, a long-term marginal distribution of $\varepsilon_{IE\omega}$ for each catchment was build. For a large majority of catchments, the long-term median $\varepsilon_{IE\omega}$ remains very close to zero. More specifically, ~50 % of all study catchments are characterized by a median deviation $\varepsilon_{IE\omega}$ that does not exceed ± 0.015 and ~ 70 % by a median within the interval ± 0.025 (Fig. 8a). Depending on the time stability of the *j* individual distributions in a catchment, the spread of annual deviations around these medians showed more variable pattern. Overall, for > 50 % of the study catchments, the IQR of annual deviations remained below 0.08 and for ~ 70 % below 0.10 (Fig. 8b). While catchments with "Stable" distributions exhibit in general a rather narrow spread with an average IQR ~ 0.08, catchments with distributions tagged as "Variable" feature a bit

wider spread with average IQR ~ 0.10, while still centring closely around zero. This can also clearly be seen by the selected examples in Fig. 6. The medians of the marginal distributions of the Chemung and Lee Rivers, both tagged as "Stable", are ~0.006 and ~0.004 respectively, with narrow IQRs of 0.062 and 0.094 (Fig. 6c,f). In contrast, while also featuring a marginal distribution with a median deviation $\varepsilon_{IE\omega}$ ~ 0.021, the Sava River catchment (Fig. 6i), tagged as "Variable", is characterized by a considerably wider scatter of the annual deviations around the median, as evident by the higher IQR of ~0.113. Three additional illustrative examples of well-known river basins are presented in Fig. S5. In contrast, the analysis, which uses the earliest available period as a fixed base line, shows an increase in the number of "Stable" catchments along with a slightly higher median $\varepsilon_{IE\omega}$ values. Further details are provided in the Supplement (Fig. S6a-f).

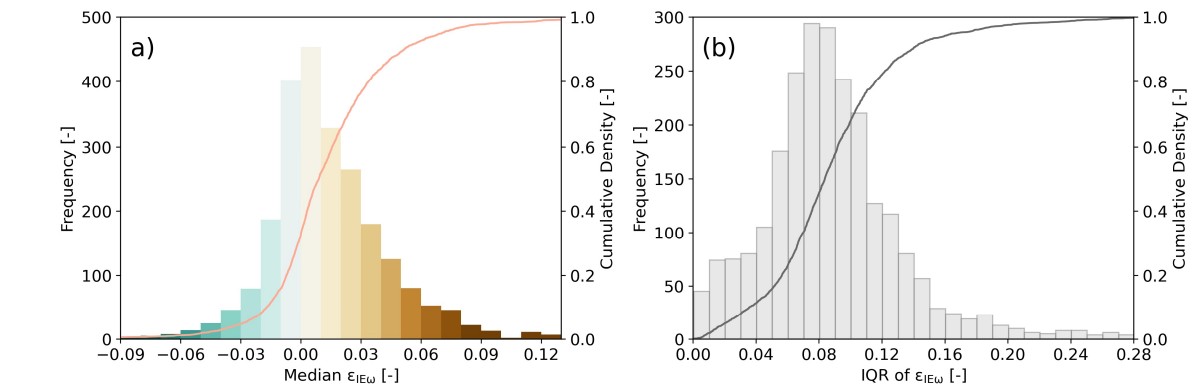

**Figure 8: Visualization of long-term a) Median $\varepsilon_{IE\omega}$ and b) Interquartile Range (IQR) of $\varepsilon_{IE\omega}$ for aggregated long-term marginal distribution of $\varepsilon_{IE\omega}$ across all catchments (2387) along with the corresponding Cumulative Distribution Function (CDF). The varying colour palette in Fig. 8a aligns with the palette used in Fig. 7b to maintain consistency. In Fig. 8b, a uniform colour is used since IQR values are all positive.**

Overall, it can be observed that median deviations $\varepsilon_{IE\omega}$ close to zero are dominant globally, with no obvious spatial clustering of more pronounced deviations (Fig. 7b). However, it can also be seen that there is some geographic grouping in the direction, i.e. the sign, of the median $\varepsilon_{IE\omega}$. While for many catchments in the central US and southern Brazil median deviations are negative, i.e. $\varepsilon_{IE\omega} < 0$, the rest of the study catchments globally are dominated by $\varepsilon_{IE\omega} > 0$. Overall, median deviations from the expected parametric Budyko curve resulted in regionally distinct relative changes in $Q$ across the studied catchments with around ~68 % of the catchments exhibiting changes $\Delta Q$ of less than $\pm 10$ % (Fig. 7c). However, catchments in some regions, notably in central US and Southern Africa, can be characterized by $\Delta Q$ exceeding $\pm 25$ %. Overall, the results indicate that catchments in more arid regions ($I_A > 2$) are particularly susceptible to relative changes in discharge as compared to more humid regions (inset Fig. 7c).

For a more regional evaluation, the yearly $\varepsilon_{IE\omega}$ values for individual catchments were aggregated into regional marginal distributions of $\varepsilon_{IE\omega}$ stratified according to the long-term mean aridity index $I_A$ and varied latitude bands (Fig. 9a).

These regional distributions capture the variability of yearly $\varepsilon_{IE\omega}$ across regions with the median $\varepsilon_{IE\omega}$ serving as a robust measure of central tendency. The general pattern found across most regions with available data are broadly consistent. 16 out of 20 regions are characterized by median deviations $\varepsilon_{IE\omega}$ that do not exceed $\pm\,0.02$. Similarly, no consistent directional pattern in the magnitude of regional median $\varepsilon_{IE\omega}$ could be identified either (Fig. 9b). For higher latitude regions beyond $\pm30^\circ$, the minor fluctuations in median $\varepsilon_{IE\omega}$ bear no evidence for a relationship with $I_A$. On the other hand, the data suggest that the spread around the regional medians consistently decreases with increasing $I_A$ across all latitude bands except $50^\circ$ N–$90^\circ$ N band as shown by the sequence of IQR in Fig. 9c. This indicates that catchments in more humid regions across the study domain are subject to more pronounced annual water storage fluctuations.

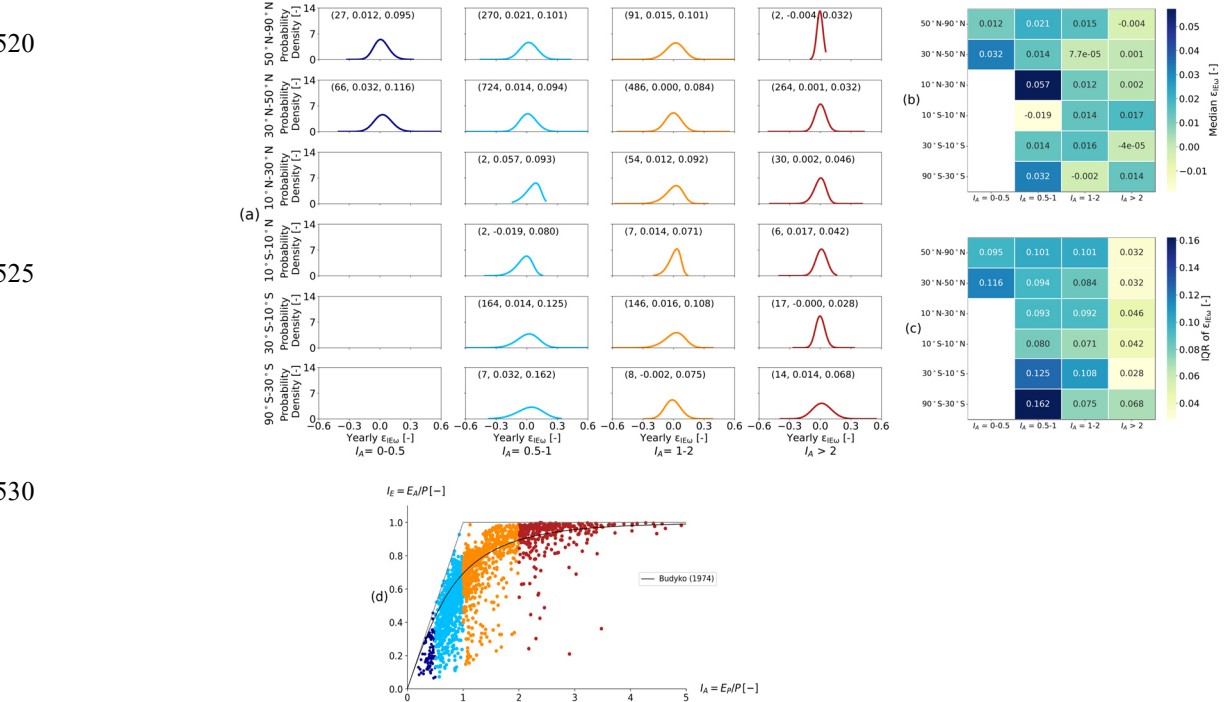

Figure 9: a) Regional marginal distributions of $\varepsilon_{IE\omega}$ for defined latitude and $I_A$ bins. The three numerical values in small brackets at the top of each panel presents number of catchments in that category, long-term median $\varepsilon_{IE\omega}$ and IQR value of $\varepsilon_{IE\omega}$ respectively, b) and c) presents variation of median and IQR values of $\varepsilon_{IE\omega}$ for the regional marginal distribution of $\varepsilon_{IE\omega}$, d) Long-term position (1901-2015) of catchments in Budyko space. The colour of the dots corresponds to the regional marginal distributions of $\varepsilon_{IE\omega}$ for the corresponding $I_A$ bin.

The parameters of the fitted parametric regional and catchment-specific marginal distributions together with the associated predictive robustness flags, as defined by the time stability of the j individual distributions for each catchment are provided in Supplementary data (https://doi.org/10.5281/zenodo.14060926) and can be used, depending on their robustness flag, to estimate $I_{E,t} = I_{E,i} + \varepsilon_{IE\omega}$ for a catchment under future hydro-climatic conditions.

**3.5 Sensitivity of marginal distributions of deviations $\varepsilon_{IE\omega}$ to the choice of 20-year averaging window (Step 6)**

The moving window analysis to quantify the sensitivity of the marginal distributions of $\varepsilon_{IE\omega}$ resulted in 20 individual, yet not uncorrelated, marginal distributions for each study catchment. Through this approach, we observed that the aggregated marginal distributions of $\varepsilon_{IE\omega}$, may indeed be subject to differences. The magnitudes of the fluctuations vary between catchments, but remain in general rather minor.

The Chemung and Lee Rivers are two examples for a very low sensitivity of the marginal distributions of $\varepsilon_{IE\omega}$ to the choice of time periods. The differences of the medians of the two most extreme marginal distributions does not exceed $\sim 0.008$ for the Chemung (Fig. 10a) and 90 % of the medians of the 20 marginal distributions fall within an interval of merely 0.01. In addition, the distributions maintain comparable shapes. A similar behaviour was observed for the Lee River (Fig. 10b), with the medians of the two most extreme distributions differing only by $\sim 0.014$. In this case, 75 % of the medians are observed within an interval of 0.01.

However, in case of catchments, that are tagged as "Variable", "Alternating" or "Shift", the difference in medians of the two extreme marginal distributions is observed to be increased. For the Sava River catchment (Fig. 10c), tagged as "Variable", the difference between the two extreme marginal distributions is $\sim 0.018$ with 75 % of the medians within an interval of 0.032. For Kaituna River (Fig. 10d), tagged as "Alternating", the difference between the medians of the two extreme marginal distributions is quite large with a value of 0.060. For 15 out of the 20 marginal distributions, the medians are found to fall within the range of 0.046. A similar pattern for Zschopau River (Fig. 10e), tagged as "Shift", is observed with a median difference for two extreme marginal distributions to be 0.029.

Overall, it was found that for 78 % of the study catchments 15 out of 20 time windows, i.e. 75 %, feature median deviations $\varepsilon_{IE\omega}$ within an interval of $\pm 0.035$. This further suggests that, although some sensitivity to the choice of time period can occur, the magnitude of the fluctuations remains rather minor for a large majority of catchments.

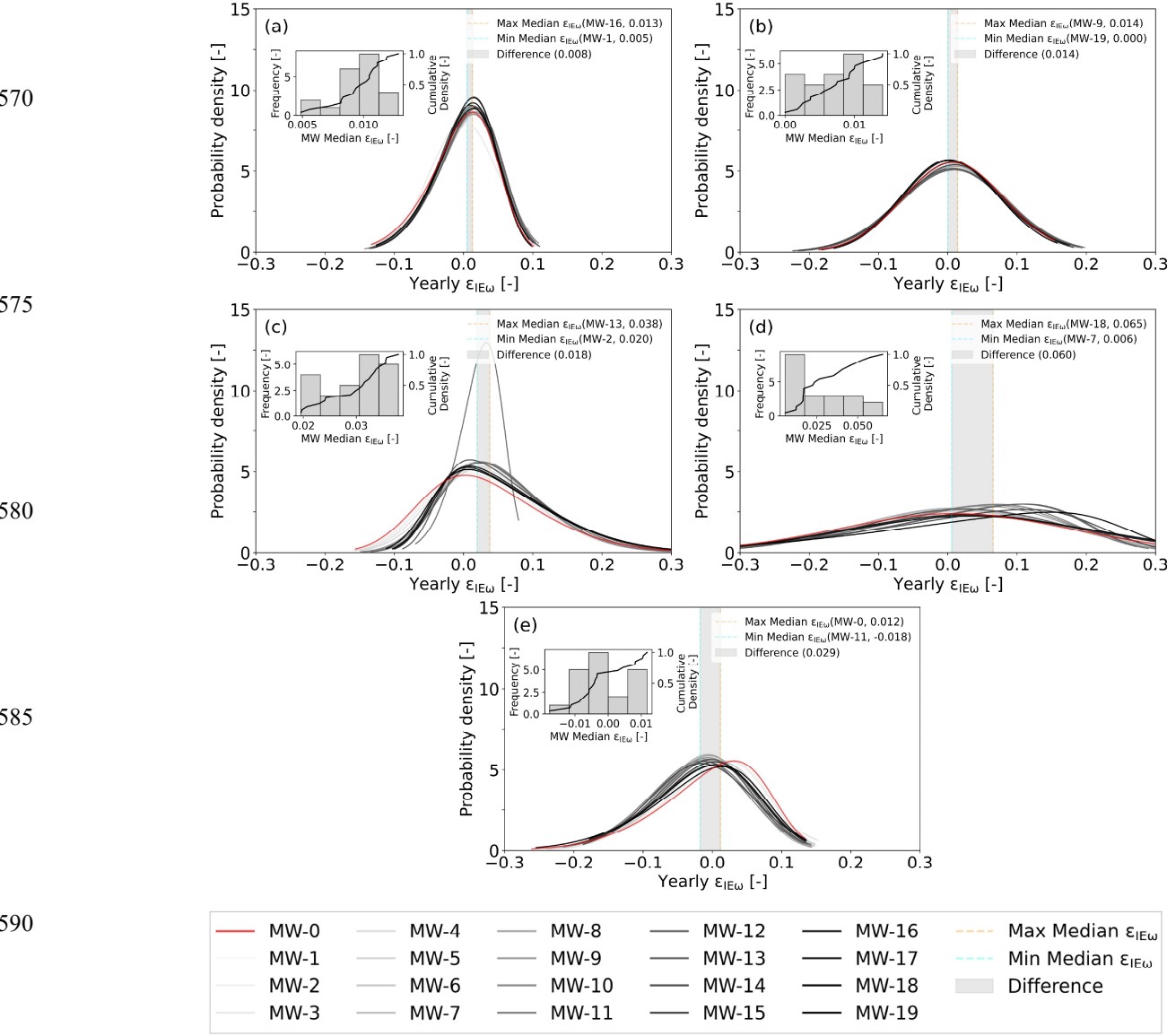

**Figure 10: Aggregated marginal distributions of $\varepsilon_{IE\omega}$ for 20 moving window time periods for five example catchments: a) Chemung River, b) Lee River, c) Sava River, d) Kaituna River, and e) Zschopau River. The red line represents the original marginal distribution of $\varepsilon_{IE\omega}$. The orange and aqua-coloured dotted lines depict the maximum and minimum median $\varepsilon_{IE\omega}$ values corresponding to their respective moving window time periods. The grey shaded area visually portrays the difference between the extreme maximum and minimum median $\varepsilon_{IE\omega}$ values across the moving window time periods.**

600         Similarly, the distribution of the median $\varepsilon_{IE\omega}$ of all study catchments, i.e. Fig. 7b, remains rather stable when evaluated over the 20 subsequent moving windows, as shown in Fig. 11, with neither the medians nor the spread of the distributions experiencing marked variations. Although lumping the medians of all catchments into one distribution may conceal fluctuations between moving windows of individual catchments, it nevertheless allows the observation that there is no systematic larger scale effect.

605

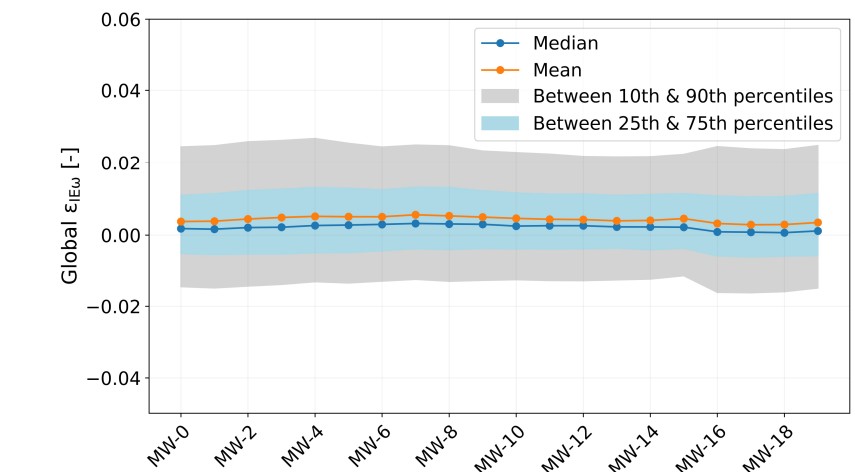

**Figure 11: Variation of global long-term median, mean, 10th & 90th percentiles and 25th & 75th percentiles of $\varepsilon_{IE\omega}$ for all of the catchments with respect to each moving window.**

## 4 Discussion

Our analysis revealed that most study catchments underwent continuous multi-decadal hydro-climatic fluctuations throughout the 20th century (Fig. 4 & Fig. S1). Notably, these fluctuations were largely consistent across the different temporal stability categories. Unlike previous studies comparing only two time periods (Jaramillo and Destouni, 2014), here the higher temporal resolution into with up to five 20-year periods, showed that these fluctuations were not one-directional, with the first half of the century trending towards higher aridity and the latter half towards increased humidity, suggesting cyclic behaviour

over longer time scales.

        In alignment with previous studies (Berghuijs and Woods, 2016; Jaramillo et al., 2018; Jaramillo et al., 2022; Reaver et al., 2022), our analysis suggests that following disturbances and thus changes in $I_A$, catchments do not necessarily and strictly follow their specific parametric Budyko curves as defined by parameter $\omega$. In our analysis we found that the general magnitudes of the median deviations $\varepsilon_{IE\omega}$ across all study catchments throughout the 20th century are very minor with median $\varepsilon_{IE\omega} \leq \pm 0.015$

for 50 % and $\leq \pm 0.025$ for 70 % of the catchments. This corresponds well with the results of Jaramillo and Destouni (2014)

and Jaramillo et al. (2018), who estimated over two multi-decade periods absolute mean deviations from the expected $I_E$ of $\varepsilon_{IE\omega} \sim 0.01$–$0.02$, for different regions in the world, based on an analysis of several hundred catchments.

Based on annual water balance data of ~400 catchments in the United States, Berghuijs and Woods (2016) reported an average difference of around 28 % between the spatial and the temporal sensitivity of $I_E$ to changes in $I_A$. However, a back-of-the-envelope calculation assuming an average $\omega = 2.6$ (Greve et al., 2015) suggests that even with a pronounced shift in aridity of $\Delta I_A = 0.2$ (Jaramillo and Destouni, 2014) such a 28 % difference in sensitivity leads to only minor absolute deviations from the expected $I_E$ with $\varepsilon_{IE\omega} \sim 0.01$–$0.04$ (4–8 %) for regions with the most common $I_A = 0.5$–$2.5$, which broadly corresponds with the results of our study. In contrast, (Reaver et al., 2022), using data from ~700 CAMELS US and UK catchments, provided a detailed and exhaustive analysis of possible temporal trajectories through the Budyko space over several decades. They report the mean of all study catchments' *maximum* relative deviations of the actually observed, empirical values $I_{E,o}$ from the predicted values of $I_E$ by catchment-specific curves with $\varepsilon_{IE\omega,max} = I_{E,o,max} - I_{E,max} \sim 26$ %. However, that mean value of all catchment *maxima* is strongly biased by a few rather extreme outliers in their analysis and the vast majority of their study catchments (>650 out of 728) exhibits much lower errors, with a median maximum deviation of $\varepsilon_{IE\omega,max} \sim 9$ % (see Fig. 3 in Reaver et al. (2022). It may thus prove more informative to interpret the results of Reaver et al. (2022) based on the mean instead of the maximum deviations as these average conditions do almost certainly occur more frequently. Doing so, it is plausible to assume that the deviations $\varepsilon_{IE\omega}$ will be considerably lower than the maximum $\varepsilon_{IE\omega,max} \sim 9$ % and potentially closer to the range of 4–8 % estimated above and thus overall consistent with the results of our analysis.

However, we also note that these minor deviations may have different practical implications in different climates (Fig. 7c). For example, in a humid catchment with $I_A = 0.5$ (e.g. mean annual $P = 2000$ mm year$^{-1}$, $E_P = 1000$ mm year$^{-1}$ and $Q = 1120$ mm year$^{-1}$), a deviation of $\varepsilon_{IE\omega} = 0.02$ results in $\Delta Q \sim 40$ mm year$^{-1}$, equivalent to merely 3 % of water yield, which hardly affects water supply. In contrast, the practical effects are more pronounced in arid environments. In a typical catchment with $I_A = 2$ (e.g. $P = 500$ mm year$^{-1}$, $E_P = 1000$ mm year$^{-1}$, $Q = 60$ mm year$^{-1}$), the same deviation will lead to a $\Delta Q \sim 10$ mm year$^{-1}$, equivalent to ~15 % of the available water yield, and thus have considerable higher relevance for water resources planning. For such environments, a robust quantification of expected deviations may thus prove beneficial for future estimates of water resources availability.

Despite some spatial clustering, the deviations $\varepsilon_{IE\omega}$ from the expected parametric Budyko curves do not exhibit any clear and unambiguous relationships with several climatic variables (Fig. S7). The detailed processes and reasons underlying the deviations thus remain so far unknown and may be assumed to be manifold and to vary depending on the characteristics of specific sites. In any case it is plausible to assume that the reasons are a combination of factors, including amongst others changes in precipitation volumes, seasonality and phase, changes in atmospheric water demand, changes in land cover, human interventions, such as reservoir operation or irrigation, but also violations of the assumption that $dS/dt \sim 0$ over the 20-year time periods (Han et al., 2020) and other uncertainties in the available data (Beven, 2016). Note, that a detailed exploration of this issue is beyond the scope of this paper.

To our knowledge, this is the first study to quantify the evolution of median $\varepsilon_{IE\omega}$ over multiple time periods. This allowed to build distributions to predict future $\varepsilon_{IE\omega}$ based on historic data, together with an indicative robustness flag, describing their temporal stability and thus their suitability to predict $\varepsilon_{IE\omega}$ under future hydro-climatic conditions. It was found that, globally, median $\varepsilon_{IE\omega}$ does not only remain minor but, perhaps even more importantly, also rather stable over time. For ~70 % of the study catchments the annual distributions of $\varepsilon_{IE\omega}$, and thus also their 20-year medians, were classified as "Stable". In other words, the available data suggest that over multiple 20-year periods in the past century the samples of annual deviations originate from the same distribution. This further allows some confidence to plausibly assume that $\varepsilon_{IE\omega}$ and the associated $I_E$ under projected future hydro-climatic conditions can, at least for several decades, be robustly predicted based on these distributions. However, it is important to note that the 20-year time periods used in this study, while effective for medium-term projections, may limit the ability to make long-term climate projections.

Further 19 % of catchments were classified as "Variable" as their distributions of annual deviations for the individual 20-year periods exhibit some variability. Despite this, there is no indicative evidence to link this variability to alternating fluctuations or systematic, one-directional shifts and thus to quantifiable deterministic processes. In this case, the fluctuations can be assumed to be arbitrarily variable, allowing the aggregation of a marginal distribution that reflects all available past knowledge. Although the uncertainty of that distribution may often exceed that of "Stable" catchments, resulting in somewhat lower predictive power (Montanari and Koutsoyiannis, 2014), it is reasonable to assume that $\varepsilon_{IE\omega}$ remains predictable. The fitted parametric marginal distributions of catchments tagged as "Stable" and "Variable" can be directly used to sample distributions of future annual $\varepsilon_{IE\omega}$ and to estimate the average $\varepsilon_{IE\omega}$ for that future period from the expected future $I_E$ based on $\omega$ of the past 20-year period.

For catchments tagged as "Alternating" or "Shift", the above assumption may be too optimistic. Although the sample size characterizing the evolution of $\varepsilon_{IE\omega}$ over the study period is with a maximum of j = 4 pairs of 20-year periods very small and thus no meaningful formal statistical tests could be executed, the data do not rule out the possibility that $\varepsilon_{IE\omega}$ in these catchments is characterized by alternating or shift-like behaviours. In other words, $\varepsilon_{IE\omega}$ may not be sampled from different distributions that change arbitrarily over time, but from distributions that (here) depend either on $I_E$ of the preceding time period or systematically increase or decrease over time. In these cases, the aggregated marginal distribution may produce spurious predictions of $\varepsilon_{IE\omega}$.

For catchments tagged as "Alternating", the user may instead want to consider to construct and use a conditional distribution in the form of $\varepsilon_{IE\omega|i}$, i.e. a distribution of $\varepsilon_{IE\omega}$ given the position $I_{E,i}$, for more reliable estimates. However, note that the limited data available for a maximum of four pairs of time periods, poses a practical complication to construct a meaningful conditional distribution $\varepsilon_{IE|\omega i}$, which is necessary to infer $\varepsilon_{IE|\omega i}$. Alternatively, the user can decide to base predictions only on basis of the $\varepsilon_{IE\omega}$ distribution of the last available time period to avoid the use of the marginal distribution (Montanari and Koutsoyiannis, 2014). For predictions in catchments with suspected presence of a systematic shift, tagged as "Shift", users may choose to extrapolate the fitted distribution parameters of the individual pairs of periods to account for their shifts over

time. However, here the reliability of this will depend on the strength of the individual relationship over the past and needs to be evaluated on a case-to-case basis as both categories are likely to lead to rather unreliable future estimates.

Additionally, empirical models like the Budyko framework have inherent weakness in dealing with previously unseen changes of the underlying distribution of a specific variable (here: $\varepsilon_{IE\omega}$). Our classification of catchments into "Stable", "Variable", "Alternating" and "Shift" categories aims to capture varying levels of sensitivity to changes in underlying distributions. Catchments classified as "Alternating" or "Shift", are more likely to have experienced large changes in the underlying distributions and may thus remain sensitive to future changes, making empirical model less robust for predictions in these cases. Conversely, "Stable" and "Variable" catchments exhibited much less sensitivity to past climatic variability. In the absence of statistical evidence for changing distributions, it is reasonable to assume that they remain relatively insensitive to change in the near future, allowing empirical models to provide plausible predictions.

It is important to note that approximately 89 % of the study catchments are either "Stable" or "Variable" and with only a small minority (~11 %) exhibiting "Alternating" or "Shift" behaviour. This predominance of "Stable" or "Variable" catchments supports the broad applicability of the Budyko framework for predictive purposes. Although there is no clear spatial pattern, the regional distributions of $\varepsilon_{IE\omega}$ remain, with medians of $\sim 0 - 0.02$ (Fig.9a), broadly consistent with the global distribution (Fig.8a) but also with each other across most spatial and climatic classes. This indeed suggests that the overall pattern is rather homogenous, and regional effects remain limited, making probabilistic predictions feasible in the absence of a deterministic description (Montanari and Koutsoyiannis, 2014). Thus, the presented distributions (Figs. 8a,9a) are in the absence of further information useful to quantitatively estimate the uncertainty for any specific catchment based on past information in a probabilistic way. However, caution is advised for out-of-sample catchments, where the assumption of stationarity may lead to less reliable predictions, as the framework cannot take into account systematic shifts or alternating behaviour.

Despite the challenges associated with catchments classified as "Alternating" and "Shift" the Budyko framework remains useful for identifying human-driven changes to the water cycle. Although many catchments showed only minor deviations, these deviations are key for recognizing drivers of change. Categorizing catchments into "Stable", "Variable", "Alternating" and "Shift" can guide targeted future research. For example, catchments in the "Alternating" and "Shift" categories may in the past either have been subject to more substantial human interference than those in the other categories or they may be more sensitive to human-induced changes. Further investigations into the drivers of these deviations may strengthen our understanding of how human-induced changes influence catchments responses differently in different environments.

It was further found that the choice of a specific 20-year window can indeed lead to fluctuations in the distributions of $\varepsilon_{IE\omega}$. However, the magnitude of these fluctuations remains rather limited for the vast majority of catchments. To avoid misinterpretations, we have therefore added the IQR of median $\varepsilon_{IE\omega}$ from the 20 individual moving windows as additional robustness flag for each catchment in Supplementary data downloadable from Zenodo repository

(https://doi.org/10.5281/zenodo.14060926). Lower IQR then indicate lower sensitivity to the choice of the 20-year window and thus a higher robustness of the marginal distribution for predictions of $\varepsilon_{IE\omega}$ under future conditions.

       A complete list of the parameters and robustness flags of the individual 20-year distributions as well as of the local aggregated marginal distributions with associated changes in $Q$ for each of the 2387 study catchments, but also of the regional distributions as stratified by latitude and $I_A$ are provided in the Supplementary data (https://doi.org/10.5281/zenodo.14060926).

These distributions of annual $\varepsilon_{IE\omega}$ can be directly used to predict the median $\varepsilon_{IE\omega}$ under future conditions locally in these catchments or regionally by sampling over 20 projected future years.

## 5 Conclusions

       Based on up to 100 years of hydro-climatic and streamflow data for 2387 river catchments world-wide we have here tested whether catchments follow their specific parametric Budyko curves as defined by parameter $\omega$ over multiple 20-year

periods throughout the 20[th] century.

We have found that:

(1)   62 % of the catchments do not significantly deviate from their expected parametric Budyko curves, although minor deviations were still observed. However, this also entails that a fraction of 38 % does indeed deviate.

(2)   The overall magnitude of deviations is minor. For ~70 % of the catchments the median deviations do not exceed $\varepsilon_{IE\omega} = \pm$

0.025, which is equivalent to ~ 1–4 %, depending on $I_E$. These median $\varepsilon_{IE\omega}$, when expressed as relative changes in Q, result in less than a ±10 % change in discharge for most catchments.

(3)   For 89 % of the study catchments, $\varepsilon_{IE\omega}$ can be considered highly or at least moderately well predictable based on historical data, as distributions of $\varepsilon_{IE\omega}$ in the past were shown to be stable over multiple time periods or characterized by variable fluctuations. The framework works well for most catchments; however, for out-of-sample catchments showing systematic

shifts or alternating behaviour, additional analysis may be required.

       The above implies that while catchments indeed may not strictly follow their parametric Budyko curves, as defined by parameter $\omega$, the deviations remain in general minor and predictable. The latter is of particular importance for catchments in water-limited regions, where already small deviations can considerably affect available water supply and where robust predictions of these deviations are instrumental for effective future water resources planning and management.


*Data availability.* Daily precipitation and temperature data were acquired via the GSWP-3 dataset, accessible at https://data.isimip.org/10.48364/ISIMIP.886955 (Lange, 2020). GSIM discharge data was obtained from https://doi.pangaea.de/10.1594/PANGAEA.887477 (Do et al., 2018) and https://doi.pangaea.de/10.1594/PANGAEA.887470 (Gudmundsson et al., 2018). Supplementary data containing topographic and climatic characteristics, parameters for fitted 20-

year parametric distributions of deviations, robustness flags indicating the temporal stability of aggregated marginal distributions, and associated changes in $Q$ across 2387 study catchments is available at https://doi.org/10.5281/zenodo.14060926. Regional parameters for fitted parametric marginal distributions are also provided.

*Author contributions.* MI conceptualized the study, conducted formal analysis, and prepared the manuscript with input from MCG, RE and MH.

*Competing interests.* MCG and MH are members of the Editorial Board of HESS. The authors have no other competing interests to declare.


*Financial support.* This research has been supported by Higher Education Commission of Pakistan (HEC) and TU Delft.

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
