# Peer review of "Catchments do not strictly follow Budyko curves over multiple decades but deviations are minor and predictable"

_Hydrology and Earth System Sciences, 2024_

## Referee Comment (RC2)

**General comments**

After a thorough read of the article "Catchments do not strictly follow Budyko curves over multiple decades but deviations are minor and predictable" by Ibrahim et al., I can see the amount of work and understand the main arguments of the authors. The goal is to assess the predictive power of the parametric Budyko curves, usually considered as not suitable for climate projections since they rely on a semi-empirical parameter, and the lack of physical explanation behind it questions whether fixing it to project future behaviours of catchments is pertinent. The authors show that over most of the catchments studied, from one 20-year period to the next, the distribution of deviations to the predictive curve is minimum and stable. This leads them to conclude that the Budyko framework can be used for projections under a changing climate, just considering a stable distribution of deviation around the curve as a shape of uncertainty.

The article is well written, well-illustrated and well integrated into the current literature. However, I am not sure every steps of the method are pertinent and I am not fully convinced by the conclusions drawn and how new the results are. The method compares successive periods of 20 years. The method stays pertinent when looking at a 20-year period and looking whether or not the median deviation from the curve can be considered different from zero or not (step 2). Therefore, the conclusions can only be applied to argue that the Budyko framework can be used for 20-years projections, which is rarely the temporality used for climate projections.

The method also compares successive deviation distribution, for instance to define "stable" catchments as catchments for which the deviation to the curve from one 20-year period to the next has no specific direction. However, if I understood correctly, each distribution of deviation to the curve for each 20-year period is calculated around a different curve (with the actualised parameter fitted over the previous 20-year period). Then, what if there is a trend in this parameter? I understand it is not possible to evaluate such a trend significantly due to the length of the data but it would invalidate the comparison of the successive distributions. Why not use the same curve for all periods and look if the distribution around the curve changes over time? Could the successive fit over the 20-year sliding time periods be used here to assess trends?

The authors argue that the distribution around the curve is just a natural variation around the curve ("stable" catchments) or due to regular climatic cycles ("variable" catchments). However, not all catchments fit in these categories, and since there seems to be no homogeneity in the spatial distribution or climatic characteristics of these catchments, it undermines the conclusion that the framework can be used for prediction in most catchments. It is not a generality, since such a study would need to be lead first in a catchment to check that it fits in a "stable" distribution, and whether or not it will seems arbitrary.

I feel the results would benefit from a different presentation, to help show their impact. As briefly presented on the discussion, I feel it would be more pertinent to express the deviation to the curve by how much it shifts the predicted aridity or discharge (%), rather than present changes in an abstract parameter. The impact of the shift in the parameter is different depending on the aridity of the catchment, which could be interesting to analyse and could shed the results in a different light.

Furthermore, with raw values of the shift in the parameter, it is difficult to understand whether it is a negligible change or not, as argued in the conclusion. Having more understandable orders of magnitude of this shift and the associated uncertainty would help argue that there is a potential in using a parametric equation for projection, with an inevitable associated uncertainty, which could be not be wider than the uncertainty associated to climate projection or to physical-based models. This

is however still not a very new argument, and should be made with an understanding of the counter-arguments, that we are never sure that empirical models will respond reasonably when faced with unprecedented changes in climate. I believe this study would be interesting in that regard, as, if it doesn't introduce completely new concepts, it has a broader perspectives and a more targeted objective to quantify the uncertainty associated to the deviation from the parametric curve for a catchment in the Budyko framework. It would benefit from being formulated as such.

**Specific comments**

Abstract, l11: I think "behaviour" is not the right term. You consider in your study parametric curves, where the parameter is generally considered to represent the specific behaviour of a catchment. A move along the curve is supposed to represent the changes in the catchment responses under a changing climate but with a fixed behaviour.

L176-179: Here your two sentences are contradictory. If I understood correctly, for each 20-year sub-period, you fitted the curve to the set of n=20 values, not to the 20-year average directly. Therefore, you need to change the first sentence of that paragraph which says the exact opposite.

I really like Figure 3, it helps understanding the steps of the method.

Paragraph 3.1: I am not sure I understand the pertinence of that part of the results. Is there a point in comparing the changes in climate variables at the global scale? Would it not be more pertinent to look at these changes in different groups of catchments, for instance looking to see if they relate to the categories of "stable", "variable", "alternating" or "shifting" catchments? Or geographically?

L535: You make the argument here that "the spread around the regional medians consistently decreases with increasing IA across all latitude bands " and therefore that "catchments in more humid regions across the study 535 domain are subject to more pronounced annual water storage fluctuations". However, as you say yourself, the impact of the shift is different depending on the aridity of the catchment. Here this argument would beneficiate from presenting relative changes in discharge or IE rather than changes in the parameter.

**Technical comments**

L50: The sentence could be reformulated. Maybe the word "described" is unnecessary.

L80: This sentence is also a little awkward. Especially the last part. Maybe separate it in two sentences.

L257: sentence unclear. Maybe do two sentences: "To do so, for each catchment the up to j = 4 distributions of deviations $\varepsilon IE\Delta j$ from expected $IE,i+1$ between subsequent time periods were compared and analysed for their changes over time. We have followed three sub-steps."

L325: "Combined this led to …" is an awkward sentence.

L329-330: I do not understand this comment.

L360: Supplementary material should not be cited before figures from the main article in a given paragraph. Otherwise why not include it?

---

## Author Comment (AC1)

**Reviewer 1**

We would like to thank the referee for the very constructive comments. We greatly appreciate the time and effort taken in thoroughly reviewing our manuscript and for providing valuable and insightful perspectives on our research. We will carefully consider these comments in the revision of the manuscript.

We have separated the different comments (shown in italic) and provide our replies below. Text in the original manuscript is shown in '*italic*' and revised text in '**bold**'. Line numbers mentioned in our reply refer to the original manuscript version.

*Reviewer comment:*

*I commend the authors for their manuscript "Catchments do not strictly follow Budyko curves over multiple decades, but deviations are minor and predictable". The hypothesis that the manuscript aims to test is that changes in trajectories in Budyko space are unpredictable, which is in itself a fundamental question in studies dealing with the Budyko framework. The probabilistic approach used to test the hypothesis is elegant, brings some clarity, and puts in context the different recent results of other studies. I enjoyed reading the manuscript, from the introduction to the conclusions. Their finding is also comforting for the field. I also appreciate the reflections on the latest research on the matter.*

Reply:

We are thankful for the encouraging remarks made by the reviewer on our manuscript. We are glad that the reviewer found our approach and reflections valuable to the field.

*Reviewer comment:*

*I have some suggestions for improvement below, but, in general, I have a rather positive perspective of the manuscript in terms of the scientific method, knowledge gap identification, novelty, approach, and implications. My only disappointment is the lack of exploration of the reason behind the shifting, variable, or alternating nature of some catchments, although the authors explicitly state that this is not the study's objective. Although not the sole aim of the manuscript, it would be indeed interesting to get some potential explanations for the different groups of Table 2. Under what conditions or drivers can catchments shift, variate or alternate?*

Reply:

We highly appreciate the positive feedback and thoughtful suggestions. Although it is indeed not the primary scope of the paper, we acknowledge that exploring the reasons behind the shifting, variable, or alternating nature of catchments would indeed be a valuable and interesting addition.

We examined the example catchments shown in Fig. 6 (in the manuscript) to explore potential factors influencing the fluctuations.

For the Sava River, classified here as variable, previous work by Levi et al. (2015) suggests that the relatively wide range of $\varepsilon_{IE\omega}$ fluctuations (IQR ~ 0.11; Fig.6i) can be largely attributed to hydropower developments and the associated changes in hydropower production levels, which disrupt natural flow regimes by increasing runoff during high demand and altering seasonal flow patterns (RenÖFÄLt et al., 2009; Lee et al., 2023).

In the case of the Kaituna River (Fig.6k), the pronounced alternating behaviour of the $\varepsilon_{IE\omega}$ fluctuations between -0.115 and 0.198, could not be readily explained by factors such as land use changes as estimated from the *Hilda+* gridded land cover product (Winkler et al., 2021), seasonality index of liquid precipitation, Parde Coefficients or median rainfall intensity (Fig. R1a,d,g-h). This suggests that some additional drivers, or a combination of drivers, might be influencing the catchment alternating behaviour.

[Figure]

**Figure R1: Land use changes (a-c), Parde Coefficients (d-f), seasonality indices (g), and median rainfall intensity (h) for three catchments (Kaituna, Thames, and Zschopau) across five 20-year periods (T₁–T₅).**

In addition, we examined the Thames River in the UK, which also exhibited an alternating sequence of negative $\varepsilon_{IE\omega}$, i.e. reduced evaporation, and positive $\varepsilon_{IE\omega}$, i.e. increased evaporation (Fig.R2). These fluctuations were found qualitatively consistent with land use changes from *Hilda+* data (Fig. R1b). From $T_1$ to $T_2$, a ~5% decrease in forest cover likely contributed to negative $\varepsilon_{IE\omega}$. Between $T_2$ and $T_3$, the positive $\varepsilon_{IE\omega}$ correlates with a 4.3% increase in pasture and a 1.7% increase in grass/shrubland, with

negligible change in forest cover. The subsequent decrease in evaporation from $T_3$ to $T_4$ coincides with a 1% reduction in forest cover and a ~2.5% decrease in grass/shrubland. However, during the final period ($T_4$ to $T_5$), vegetation changes cannot explain the observed alternating behaviour.

The shifting behaviour of $\varepsilon_{IE\omega}$ for the Zschopau River (Fig. 6n) into one dominant direction after $\varepsilon_{IE\Delta1}$ coincides with a gradual decrease in the seasonality of liquid precipitation input (i.e. rainfall + snowmelt; Fig.R1g), combined with an increase in forest cover (*Hilda+* data) towards the end of century (Fig.R1c). Renner et al. (2014); Renner and Hauffe (2024) also reported a gradual forest recovery in the Zschopau catchment during this period, which could further contribute to the observed shift.

While some of these explorations suggests potentially plausible correlations with land use changes and seasonality of liquid precipitation, the available data is insufficient to draw robust conclusions. Given the limited data support, we chose not to include these potential explanations in the manuscript to avoid introducing overly speculative elements.

[Figure]

**Figure R2: Individual distribution of deviations ($\varepsilon_{IE\Delta1}$, $\varepsilon_{IE\Delta2}$, $\varepsilon_{IE\Delta3}$ and $\varepsilon_{IE\Delta4}$) for Thames River in the UK**

*Reviewer comment:*

*Title: The title says that most catchments deviate, but the conclusion states "62% do not significantly deviate", which is contradicting.*

Reply:

Please note that we have made the deliberate choice to formulate the title as "Catchments do not strictly follow….". This is more general than "Most catchments do not strictly follow…" and does not directly imply a majority.

Indeed, our analysis suggests that, based on Wilcoxon Signed Rank Tests, for 62% of the catchments the median deviations were not significantly different from zero. However, it is important to note that minor deviations were still observed in most catchments even though they were not classified as significant based on this specific statistical test. It would probably too naive to assume that $\varepsilon_{IE\omega} = 0$ and that catchments therefore strictly follow their curves, as also demonstrated, for example, by (Reaver et al., 2022). We therefore prefer to reflect this in the title of the paper. We believe this title conveys the key message of this research work. However, we will clarify this point in discussion part of the revised manuscript.

*Reviewer comment:*

*l. 69 Climate is not the only driver; do not forget changes in water and land use, which have been broadly found to drive the trajectory of movement in Budyko space.*

Reply:

We completely agree. We will expand the statement in lines 68-72 as follows:

"*Recently, it was also argued that catchments should not be necessarily expected to follow their long-term average, catchment specific parametric Budyko curves when subject to climatic perturbations, expressed as changes in $I_A$ (Berghuijs and Woods, 2016; Jaramillo et al., 2018; Jaramillo et al., 2022; Reaver et al., 2022). Such deviations ($\varepsilon_{IE\omega}$) from the expected parametric Budyko curve, were previously referred to as residual or landscape-driven, **indicating that many factors other than $I_A$, such as human-induced changes in water and land use (e.g., afforestation, deforestation, irrigation, reservoir construction) can and do also play a role (Donohue et al., 2007; Wang and Hejazi, 2011; Destouni et al., 2012; Sterling et al., 2012; van der Velde et al., 2014; Jaramillo and Destouni, 2015; Levi et al., 2015; Nijzink et al., 2016; Daly et al., 2019; Gan et al., 2021; Hrachowitz et al., 2021)***

*Reviewer comment:*

*Fig. 2- The use of symbols in variables became too confusing at some point. The subindices in the variables are long and have a long set of characters, as shown in Fig. 2. Maybe this can be simplified in some way. In the same way, the critical variable $\varepsilon_{IE\omega}$ is not explicitly shown in Fig. 2. There are also some inconsistencies, e.g., $\varepsilon_{IE\Delta j}$. Why "j"? I would avoid the i+1 subindex in the figure so that it agrees with the variables called in the text.*

Reply:

We agree with that observation. Given the multi-decadal periods involved, simplifying the notations is quite challenging. However, we will revisit the figure and try to make it simpler without losing clarity. We will explicitly show $\varepsilon_{IE\omega}$ in the revised figure and address any inconsistencies throughout the revised manuscript.

*Reviewer comment:*

*Fig. 5. $\varepsilon_{IE\Delta}$ does not agree with its expression in Fig. 2. This also brings confusion. Please double-check these issues across the manuscript.*

Reply:

We thank the reviewer for pointing this out. We will correct that and ensure consistency in use of symbols throughout the revised manuscript.

*Reviewer comment:*

*L. 186 Why are the time periods consecutive? I see no problem in comparing the changes from, for example, $T_1$ to $T_4$. These new permutations would give even more robustness to the statements of deviation or not deviation.*

Reply:

We do, in principle, agree and considered this approach in the initial phase of the research. However, we chose to proceed with comparing consecutive periods for the following reasons:

1. The use of the most recent 20-year period as the baseline allows to explore the temporal stability of changes. Meaning, is there a systematic pattern over time, as reflected in the classes "Alternating and Shifts" or can the fluctuations be assumed to be random, as reflected in the classes "Stable" and "Variable"? Using permutations of different periods and thus not preserving the temporal sequence would make such a distinction problematic. We believe that such a classification is necessary for the interpretation of any type of future estimates of $I_E$. For catchments in the classes "Alternating" and "Shift", historical data indicate a change of the underlying distributions which need to be considered for any type of future estimation. In the absence of further information, it is then plausible to assume that the most recent distributions are more representative as baseline for predictions.

2. In many cases, we do not have complete data for five 20-year periods across the full 100-year timeframe. As only 159 out of 2,387 catchments include the full 100-year period for $\Delta_{1\text{-}2}$, while 889 catchments have data for $\Delta_{2\text{-}3}$, and the number increase to 1916 for $\Delta_{3\text{-}4}$, and 2269 for $\Delta_{4\text{-}5}$. This distribution shows that for most catchments, only two or three 20-year periods are available, making changes to the baseline period less impactful.

In any case, we will include this in the discussion part of our revised manuscript.

*Reviewer comment:*

*L. 200 ω is both the Budyko and PDF scaling parameters, which is also confusing.*

Reply:

We thank the reviewer for this observation. In the revised manuscript, we will replace the scale parameter ω of Skew Normal Distribution with λ to avoid confusion.

*Reviewer comment:*

*Fig. 3 Mention the example of the basin you are showing here.*

Reply:

This figure is not based on real data and is intended for illustration purpose only. We will clarify this in the revised manuscript.

*Reviewer comment:*

*Fig. 6. Can you classify the catchments with the classification of Table 2? This helps understand which ones correspond to which. Also, why did you choose these catchments? I would also put the name of the catchment in the plots.*

Reply:

Excellent suggestion. In the revised manuscript, we will mention both the class and names of each catchment in the plots. The selected example catchments were chosen as they provide a good representation of the different categories used in this research.

*Reviewer comment:*

*L. 395 The answer to your finding about the Sava River may be found in Levy et al. (2015); hydropower development.*

Reply:

We thank the reviewer for pointing us to that paper. We will include this information in the discussion section of the revised manuscript.

*Reviewer comment:*

*Fig. 8 Any use for the palette change in Fig. 8a?*

Reply:

The palette in Fig. 8a differs from that in Fig. 8b to match the palette used in Fig. 7b. This was done to maintain consistency and help the reader to better understand the figures. In contrast, Fig. 8b uses a single colour as the interquartile range (IQR) values are all positive, and thus a more varied colour palette was not necessary. The choice of colour is just random. We will clarify the use of the different colour schemes in the figure caption.

*Reviewer comment:*

*Discussion: I would like to know the thoughts from the authors on the future use of the framework for identifying human modifications to the water cycle, as it has largely been used to date. Maybe some recommendations on the way forward for this goal could be included in the discussion.*

*For instance, the fact that most basins do not deviate does not necessarily mean that the Budyko framework (and the authors' approach) cannot be used to continue identifying human drivers of change. In fact, such identification relies on the deviations to recognize drivers of change. A way forward can be the categorization of the authors into stable, variable, alternating, and shifting categories and to focus analysis on some of these groups.*

Reply:

We completely agree that despite the minor deviations observed in the majority of catchments, the Budyko framework remains useful for identifying human-driven changes. Indeed, deviations are key

for recognizing these drivers of change. Categorizing catchments into "Stable", "Variable", "Alternating" and "Shift" could guide a targeted future research. For example, catchments in the "Alternating" and "Shift" categories may in the past either have been subject to more substantial human interference than those in the other categories or they may be more *sensitive* to human-induced changes. Further investigations into the drivers of these deviations could enhance our understanding of how these human-induced changes influence catchments responses differently in different environments. We will incorporate and discuss this this perspective in the revised manuscript.

*Reviewer comment:*

*Could the authors provide a list in Supplementary on the catchments that fall in each of the categories (if this is not already mentioned).*

Reply:

We thank the reviewer for this suggestion. A detailed summary table, including the list of catchments with their respective categories, is available in a separate repository available at: https://doi.org/10.5281/zenodo.10925966. We will add this to our data availability statement.

**References:**

[revised manuscript text omitted]

---

## Author Comment (AC2)

**Reviewer 2**

We would like to thank the referee for the detailed comments. We greatly appreciate the time and effort taken in thoroughly reviewing our manuscript and for providing valuable and insightful perspectives on our research. We will carefully consider these comments while revising the manuscript.

We have separated the different comments (shown in italic) and provide our replies below. Text in the original manuscript is shown in '*italic*' and revised text in '**bold**'. Line numbers mentioned in our reply refer to the original manuscript version.

*Reviewer comment:*

*After a thorough read of the article "Catchments do not strictly follow Budyko curves over multiple decades but deviations are minor and predictable" by Ibrahim et al., I can see the amount of work and understand the main arguments of the authors. The goal is to assess the predictive power of the parametric Budyko curves, usually considered as not suitable for climate projections since they rely on a semi-empirical parameter, and the lack of physical explanation behind it questions whether fixing it to project future behaviours of catchments is pertinent. The authors show that over most of the catchments studied, from one 20-year period to the next, the distribution of deviations to the predictive curve is minimum and stable. This leads them to conclude that the Budyko framework can be used for projections under a changing climate, just considering a stable distribution of deviation around the curve as a shape of uncertainty.*

Reply:

We would like to thank the reviewer for a thorough review and for highlighting the objective of our research. We appreciate the acknowledgement of our efforts to evaluate the predictive power of the parametric Budyko curves.

*Reviewer comment:*

*The article is well written, well-illustrated and well-integrated into the current literature. However, I am not sure every steps of the method are pertinent and I am not fully convinced by the conclusions drawn and how new the results are.*

Reply:

We highly appreciate the reviewer's encouraging feedback on the writing, illustration and integration of our manuscript into the current literature.

*Reviewer comment:*

*The method compares successive periods of 20 years. The method stays pertinent when looking at a 20-year period and looking whether or not the median deviation from the curve can be considered different from zero or not (step 2). Therefore, the conclusions can only be applied to argue that the Budyko framework can be used for 20-years projections, which is rarely the temporality used for climate projections.*

Reply:

We acknowledge the observation that the chosen approach is valid strictly for predictions over 20-year windows, while less so for longer-range predictions. In the early phase of the study design, we have alternatively also considered longer time windows, but eventually, deliberately decided to use windows of 20 successive years as an approach that balances the need for sufficiently long time periods to limit the effect of storage changes d$S$/d$t$ (Han et al., 2020), while preserving the temporal sequence in the data that allowed us to place each catchment into a specific category (i.e. "Stable", "Variable", "Alternating" or "Shift"). This aspect is one of the major novelties of our analysis, as it has – to our knowledge – never been analysed on global scale before. The use of fewer but longer time windows, such as ~ 30 years, as previously done by others, e.g., Destouni et al. (2012), would have considerably limited a meaningful distinction of systematic shifts from more random fluctuations over time.

We would also like to emphasize that 20-year periods are a not uncommon time-horizon for many water resources management interventions and planning, where such shorter-term predictions are often more relevant for decision-making.

However, we completely acknowledge the limitations of our choice. We will therefore add our reasoning for the 20-years window and a detailed discussion of the implications of this choice in the revised manuscript.

*Reviewer comment:*

*The method also compares successive deviation distribution, for instance to define "stable" catchments as catchments for which the deviation to the curve from one 20-year period to the next has no specific direction. However, if I understood correctly, each distribution of deviation to the curve for each 20-year period is calculated around a different curve (with the actualised parameter fitted over the previous 20-year period). Then, what if there is a trend in this parameter? I understand it is not possible to evaluate such a trend significantly due to the length of the data but it would invalidate the comparison of the successive distributions. Why not use the same curve for all periods and look if the distribution around the curve changes over time? Could the successive fit over the 20-year sliding time periods be used here to assess trends?*

Reply:

We greatly appreciate the reviewer's sharp observation and agree that a fixed baseline can provide additional insights. To explore this, we conducted the analysis using the reviewer's proposed approach by fixing the first 20-year period as the baseline and calculated distributions of deviations for the subsequent 20-year periods accordingly.

This comparison is illustrated in Figures R1 and R2 below, in which we compare the original approach with a dynamic baseline to the use of a fixed baseline for two of the example catchments of our study. For the first example catchment (Chemung River, Fig. 6a-c in the manuscript), we observed that the results from both methods were almost identical (Fig. R1a-b & R2a-b). However, for the second example catchment (Lee River Fig. 6d-f in the manuscript), the median deviations were somewhat higher when using a fixed baseline (Fig. R1c-d & R2c-d).

We also extended this analysis to all other study catchments. However, please note that we could include the full 100-year period in only 159 out of 2387 catchments. For the other catchments, we

used the oldest available 20-year period as the fixed baseline. We have found that the proportion of "Stable" catchments increased from 72% (dynamic baseline - original approach) to 84% (fixed baseline - proposed approach) (Fig. R3), suggesting that dynamic baselines are more sensitive for the detection of systematic changes, i.e. "shifting" (trends) or "alternating" behaviour. In contrast, while the number of stable catchments increased, we also observed that the median deviations of the aggregated marginal distributions of deviations were slightly higher when using a fixed baseline (Fig. R4).

Despite the fact that there are some interesting results, we still prefer to keep the temporally changing (dynamic) baseline in the main part of our analysis for the following reasons:

1. The use of the most recent period as the baseline for assessing temporal stability of catchments is particularly relevant for future predictions, as the most recent data are most likely to provide a meaningful representation of current conditions. Using the oldest period as the baseline may not reflect recent conditions and could result in misleading conclusions, in particular when the first and last periods are far apart.
2. As previously mentioned, only 159 out of 2,387 catchments include the full 100-year period for $\Delta_{1-2}$, while 889 catchments have data for $\Delta_{2-3}$, and the number increase to 1916 for $\Delta_{3-4}$, and 2269 for $\Delta_{4-5}$. This distribution shows that for most catchments, only two or three 20-year periods are available, making changes to the baseline period less impactful.
3. As the reviewer correctly pointed out, it is difficult to quantify trends in the omega parameter based on only 5 values due to the limited length of the available data records. In many cases, we are working with just 2 or 3 time periods, making it impossible to detect significant trends.

Overall, adjusting the baseline over time is more sensitive to capturing recent shifts and trends in hydrological behaviour, which helps to assign catchments to one of the four categories.

The reviewer's suggestion to use successive fits over sliding 20-year periods to assess trends is indeed interesting. However, this would introduce dependencies between the periods, as each successive period overlaps with the previous one. This, in turn, would compromise the independence of the data from the individual time periods and potentially bias the results.

Based on the above and to provide the reader with a more comprehensive view of our results, we will, as the reviewer has suggested, include the results of using a single fixed baseline (Fig. R3 and R4) in the Supplementary Material of the revised manuscript. In addition, we will include a discussion of both approaches in the revised manuscript, outlining their respective benefits and limitations.

[Figure]

**Figure R1: Comparison of individual distribution of deviations (ε$_{IEΔ1}$, ε$_{IEΔ2}$, ε$_{IEΔ3}$ and ε$_{IEΔ4}$) between the dynamic baseline (left) and fixed baseline (right) approaches for two example catchments (Chemung River and Lee River) in the Stable category**

[Figure]

**Figure R2: Comparison of long-term marginal distribution of annual deviations between the dynamic baseline (left) and fixed baseline (right) approaches for two example catchments (Chemung River and Lee River) in the Stable category**

[Figure]

**Figure R3: Comparison of temporal stability of the studied catchments using the dynamic baseline (top) and fixed baseline (bottom) approaches. Catchments highlighted with a black border represent the 5 selected examples from Fig. 6 (of the original manuscript), while those outlined in red denote three additional selected example catchments shown in the supplement (Fig. S4 of the original manuscript)**

[Figure]

**Figure R4: Comparison of long-term median $\epsilon_{IE\omega}$ values of aggregated marginal distribution of $\epsilon_{IE\omega}$ across all catchments (2387) comparing the dynamic baseline (left) and fixed baseline (right) approaches**

*Reviewer comment:*

*The authors argue that the distribution around the curve is just a natural variation around the curve ("stable" catchments) or due to regular climatic cycles ("variable" catchments). However, not all catchments fit in these categories, and since there seems to be no homogeneity in the spatial distribution or climatic characteristics of these catchments, it undermines the conclusion that the framework can be used for prediction in most catchments. It is not a generality, since such a study would need to be lead first in a catchment to check that it fits in a "stable" distribution, and whether or not it will seems arbitrary.*

Reply:

While we agree with the reviewer's observation that not all catchments fit into the categories of "Stable" or "Variable", it is important to note that these catchments ("Alternating" or "Shift") constitute only a small minority, comprising 12% of the study catchments. The remaining 88% of catchments are either "Stable" or "Variable".

As pointed out by the reviewer, there is no clear spatial pattern. The regional distributions of $\epsilon_{IE\omega}$ remain, with medians of $\sim 0 - 0.02$ (Fig.9 in the original manuscript), broadly consistent with the global distribution (Fig.8) but also with each other across most spatial and climatic classes. This indeed suggests that the overall pattern is rather homogenous, and regional/local effects remain limited. The presented distributions (Figs. 8, 9) are in the absence of further information nevertheless useful to quantitatively estimate the uncertainty for any specific catchment based on past information in a probabilistic way, as clearly pointed out by Montanari and Koutsoyiannis (2014): "*[...] If a deterministic description of the process statistics along time, applicable to future times, is not available, which implies that non-stationarity is impossible to define, the only way for making predictions is through assumptions of stationarity*". Whereby "*if a deterministic description [...] is not available*" in our cases corresponds to the impossibility to identify "shifting" and "alternating" out-of-sample catchments and "*assumption of stationarity*" corresponds in our case to the assumption that out-of-sample catchments are largely "stable" and "variable".

We will discuss these points in more detail and revise our conclusions to reflect that while the framework works well for the wide majority of catchments, it cannot take into account systematic shifts or alternating behaviour in out-of-sample catchments.

*Reviewer comment:*

*I feel the results would benefit from a different presentation, to help show their impact. As briefly presented on the discussion, I feel it would be more pertinent to express the deviation to the curve by how much it shifts the predicted aridity or discharge (%), rather than present changes in an abstract parameter. The impact of the shift in the parameter is different depending on the aridity of the catchment, which could be interesting to analyse and could shed the results in a different light.*

Reply:

We thank the reviewer for this excellent suggestion. We agree that expressing the deviation from the curve in terms of the percentage change in the evaporative index or discharge (or runoff coefficient) provides a better understanding of the impact. We have calculated the percentage change in discharge for each catchment and will include this analysis in the revised manuscript (Fig. R5). Additionally, we have analysed these changes across four different aridity classes to further highlight how the impact of the parameter shift varies with catchment aridity (inset in Fig. R5). This approach has allowed for a more meaningful presentation of the results, and it will be incorporated into the revised manuscript. This figure will be added as Figure 7c in the revised manuscript.

[Figure]

**Figure R5: Spatial distribution map of changes in Q as a result of long-term median $\epsilon_{IE\omega}$ values. Histogram and cumulative density of changes in Q and change in Q across different $I_A$ bins are presented as two insets. Catchments highlighted with a black border represent the 5 selected examples from Fig. 6 (of the original manuscript), while those outlined in red denote three additional selected example catchments shown in the supplement (Fig. S4 of the original manuscript). The boxes represent the 25$^{th}$ to 75$^{th}$ percentiles, while whiskers extend to the 10$^{th}$ and 90$^{th}$ percentiles. Diamonds denote the arithmetic mean, and outliers are not shown.**

*Reviewer comment:*

*Furthermore, with raw values of the shift in the parameter, it is difficult to understand whether it is a negligible change or not, as argued in the conclusion. Having more understandable orders of magnitude of this shift and the associated uncertainty would help argue that there is a potential in using a parametric equation for projection, with an inevitable associated uncertainty, which could be not be wider than the uncertainty associated to climate projection or to physical-based models. This is however still not a very new argument, and should be made with an understanding of the counter-arguments, that we are never sure that empirical models will respond reasonably when faced with unprecedented changes in climate. I believe this study would be interesting in that regard, as, if it doesn't introduce completely new concepts, it has a broader perspectives and a more targeted objective to quantify the uncertainty associated to the deviation from the parametric curve for a catchment in the Budyko framework. It would benefit from being formulated as such.*

Reply:

We agree that it can be difficult to interpret the relevance of change based only on raw values of the shift in the parameter. The reviewer's suggestion to show the magnitude of the shift in forms of a magnitude change such as % change in discharge is indeed very helpful and we will include this in the revised manuscript. Furthermore, we also fully agree that it is an inherent weakness of empirical models that they typically cannot deal with previously unseen changes of the underlying distribution of a specific variable. Identifying the sensitivity of catchments to such changes in underlying distributions to address exactly this issue was our main intention with the classification of the study catchments into four categories. Following both, the shorter term climatic variability but also the long-term changing climate over the past century, catchments classified as "Alternating" or "Shift", are more likely to have experienced changes in the underlying distributions and are thus plausible to remain sensitive to change in the future. For such catchments, the empirical model will thus be less robust for predictions. However, for the other categories i.e., "Stable" or "Variable", the catchments exhibited

much less sensitivity to past climatic variability. In the absence of statistical evidence for changing distributions it is thus not implausible to assume that they remain, at least for the near-future, rather insensitive to change and that the empirical models can thus provide plausible predictions. However, we explicitly acknowledge that issue remains a limiting characteristic of statistical models. We will include these points in the discussion section of the revised manuscript.

*Reviewer comment:*

*Abstract, l11: I think "behaviour" is not the right term. You consider in your study parametric curves, where the parameter is generally considered to represent the specific behaviour of a catchment. A move along the curve is supposed to represent the changes in the catchment responses under a changing climate but with a fixed behaviour.*

Reply:

Indeed. We will modify Line 11 as follows:

"**A movement along a specific Budyko curve with changes in the aridity index over time has been used as a predictor for catchment responses to changing climatic conditions**"

*Reviewer comment:*

*L176-179: Here your two sentences are contradictory. If I understood correctly, for each 20-year sub-period, you fitted the curve to the set of n=20 values, not to the 20-year average directly. Therefore, you need to change the first sentence of that paragraph which says the exact opposite.*

Reply:

We thank the reviewer for pointing this out. We will modify the lines 176-179 as follows:

"**For each catchment and each individual 20-year time period $T_i$, the catchment-specific parametric Budyko curve $I_{E,i}$ defined by parameter $\omega_i$ is obtained by fitting Eq.(2) to the set of 20 annual values of each catchment in the Budyko space, as computed from the observed water balance data**".

*Reviewer comment:*

*I really like Figure 3, it helps understanding the steps of the method.*

Reply:

We are glad that the reviewer found this figure helpful.

*Reviewer comment:*

*Paragraph 3.1: I am not sure I understand the pertinence of that part of the results. Is there a point in comparing the changes in climate variables at the global scale? Would it not be more pertinent to look at these changes in different groups of catchments, for instance looking to see if they relate to the categories of "stable", "variable", "alternating" or "shifting" catchments? Or geographically?*

Reply:

We thank the reviewer for the valuable feedback and suggestion. The rationale for showing periodic changes in climatic variables, at a global scale throughout the study period, is to provide an overall view of how the key variables that determine the Budyko coordinates ($E_A/P$ and $E_P/P$), have changed over time. This global perspective helps to understand the trends and catchments' movement within the Budyko space. However, we also agree with the suggestion to provide a more detailed description based on the catchment categories for this analysis. We have carried out the proposed analysis and will incorporate it in the revised manuscript by replacing Figure 4 with Figure R6 (see here below).

[Figure]

**Figure R6: Temporal stability category-wise mean 20-year changes in hydro-climatic variables for the studied catchments between two consecutive periods. a) Precipitation $P$, b) Temperature $T$, c) Potential evaporation $E_P$, d) Actual evaporation $E_A$, e) Aridity index $I_A$, and f) Evaporative index $I_E$. The boxes represent the 25th to 75th percentiles, while whiskers extend to the 10th and 90th percentiles. Diamonds denote the arithmetic mean, and outliers are not shown**

*Reviewer comment:*

*L535: You make the argument here that "the spread around the regional medians consistently decreases with increasing IA across all latitude bands" and therefore that "catchments in more humid regions across the study 535 domain are subject to more pronounced annual water storage fluctuations". However, as you say yourself, the impact of the shift is different depending on the aridity of the catchment. Here this argument would beneficiate from presenting relative changes in discharge or IE rather than changes in the parameter.*

Reply:

We completely agree that presenting relative changes in discharge or $I_E$ rather than changes in the parameter would provide a clearer understanding. We have conducted the analysis as suggested and

will include the results in the revised manuscript by adding Figure R5 as Figure 7c in the revised manuscript.

*L50: The sentence could be reformulated. Maybe the word "described" is unnecessary.*

Reply:

We thank the reviewer for the suggestion. We will modify Line 50 as follows:

**"The fact that the long-term water balance exhibits such a relatively consistent behaviour across a wide spectrum of hydro-climatically and physio-graphically distinct environments has led to the hypothesis that the general shape of Budyko curves emerges for natural systems in a co-evolution of climate, soil water storage and vegetation properties".**

*L80: This sentence is also a little awkward. Especially the last part. Maybe separate it in two sentences.*

Reply:

We will modify Line 80 as follows:

**"In other words, some level of deviation from the expected parametric Budyko curves is to be expected, as different time periods will never be characterized by exactly the same environmental conditions. However, the mechanistic processes that control these deviations, and thus ω, are not well understood".**

*L257: sentence unclear. Maybe do two sentences: "To do so, for each catchment the up to j = 4 distributions of deviations $\varepsilon_{IE\Delta j}$ from expected $I_{E,i+1}$ between subsequent time periods were compared and analysed for their changes over time. We have followed three sub-steps."*

Reply:

We will split the sentence into two, as the reviewer suggested. The modified sentences are as follows:

"**To do so, for each catchment the up to j = 4 distributions of deviations $\varepsilon_{IE\Delta j}$ from expected $I_{E,i+1}$ between subsequent time periods were compared and analysed for their changes over time. We have followed three sub-steps**".

*L325: "Combined this led to …" is an awkward sentence.*

Reply:

We will modify Line 325 as follows:

*"These combined factors led to slightly more arid conditions in the first half of the 20$^{th}$ century, followed by a considerable reduction of aridity index I$_A$ and thus to a shift towards somewhat more humid conditions towards the end of the century across all of the temporal stability categories (Fig. 4e), in which, on average 78% and 75% of the catchments showed decreases in I$_A$ for Δ$_{3-4}$ and Δ$_{4-5}$, respectively".*

*Reviewer comment:*

*L329-330: I do not understand this comment.*

Reply:

We have modified these lines as follows:

*"The movement of catchments in the Budyko space due to hydro-climatic changes are illustrated in form of rose diagrams in Fig. S1 (Jaramillo and Destouni, 2014). If these movements were driven only by changes in I$_A$, catchments would be expected to move within the directional range of 0 < α < 45$^o$ or 225 < α < 270$^o$ (Jaramillo et al., 2022). However, observed movement of catchments are also found in other directions, indicating deviations ε$_{IEω}$ from the expected I$_E$, as elaborated in detail in Fig. S1".*

Furthermore, Section 3.1 of the original manuscript will be revised in the light of new insights obtained after replacing Figure 4 with Figure R6.

*Reviewer comment:*

*L360: Supplementary material should not be cited before figures from the main article in a given paragraph. Otherwise why not include it?*

Reply:

We have removed the citation of supplementary material in this paragraph and inserted it after elaboration of the previous comment.

**References**

Destouni, G., Jaramillo, F. and Prieto, C. 2012. Hydroclimatic shifts driven by human water use for food and energy production. Nature Climate Change 3(3), 213-217, https://doi.org/10.1038/nclimate1719.

Han, Z., Long, D., Huang, Q., Li, X., Zhao, F. and Wang, J. 2020. Improving Reservoir Outflow Estimation for Ungauged Basins Using Satellite Observations and a Hydrological Model. Water Resources Research 56(9), https://doi.org/10.1029/2020wr027590.

Jaramillo, F. and Destouni, G. 2014. Developing water change spectra and distinguishing change drivers worldwide. Geophysical Research Letters 41(23), 8377-8386, https://doi.org/10.1002/2014gl061848.

Jaramillo, F., Piemontese, L., Berghuijs, W.R., Wang-Erlandsson, L., Greve, P. and Wang, Z. 2022. Fewer Basins Will Follow Their Budyko Curves Under Global Warming and Fossil-Fueled Development. Water Resour Res 58(8), e2021WR031825, https://doi.org/10.1029/2021WR031825.

Montanari, A. and Koutsoyiannis, D. 2014. Modeling and mitigating natural hazards: Stationarity is immortal! Water Resources Research 50(12), 9748-9756, https://doi.org/10.1002/2014wr016092.

---

## Author Response (AR1)

**Note: Adjustments in response to the comments from Reviewer 1 are highlighted in yellow, those for Reviewer 2 are in green, and common points raised by both reviewers are highlighted in pink in the revised manuscript.**

**Reviewer 1**

We would like to thank the referee for the very constructive comments. We greatly appreciate the time and effort taken in thoroughly reviewing our manuscript and for providing valuable and insightful perspectives on our research. We will carefully consider these comments in the revision of the manuscript.

We have separated the different comments (*shown in italic*) and provide our replies below. Text in the original manuscript is shown in '*italic*' and revised text in '***bold***'.

*Reviewer comment:*

*I commend the authors for their manuscript "Catchments do not strictly follow Budyko curves over multiple decades, but deviations are minor and predictable". The hypothesis that the manuscript aims to test is that changes in trajectories in Budyko space are unpredictable, which is in itself a fundamental question in studies dealing with the Budyko framework. The probabilistic approach used to test the hypothesis is elegant, brings some clarity, and puts in context the different recent results of other studies. I enjoyed reading the manuscript, from the introduction to the conclusions. Their finding is also comforting for the field. I also appreciate the reflections on the latest research on the matter.*

Reply:

We are thankful for the encouraging remarks made by the reviewer on our manuscript. We are glad that the reviewer found our approach and reflections valuable to the field.

*Reviewer comment:*

*I have some suggestions for improvement below, but, in general, I have a rather positive perspective of the manuscript in terms of the scientific method, knowledge gap identification, novelty, approach, and implications. My only disappointment is the lack of exploration of the reason behind the shifting, variable, or alternating nature of some catchments, although the authors explicitly state that this is not the study's objective. Although not the sole aim of the manuscript, it would be indeed interesting to get some potential explanations for the different groups of Table 2. Under what conditions or drivers can catchments shift, variate or alternate?*

Reply:

We highly appreciate the positive feedback and thoughtful suggestions. Although it is indeed not the primary scope of the paper, we acknowledge that exploring the reasons behind the shifting, variable, or alternating nature of catchments would indeed be a valuable and interesting addition.

We examined the example catchments shown in Fig. 6 (in the manuscript) to explore potential factors influencing the fluctuations.

For the Sava River, classified here as "Variable", previous work by Levi et al. (2015) suggests that the relatively wide range of $\varepsilon_{IE\omega}$ fluctuations (IQR ~ 0.11; Fig. 6i) can be largely attributed to hydropower developments and the associated changes in hydropower production levels, which disrupt natural flow regimes by increasing runoff during high demand and altering seasonal flow patterns (Lee et al., 2023; Renofalt et al., 2010).

In the case of the Kaituna River (Fig. 6k), the pronounced alternating behaviour of the $\varepsilon_{IE\omega}$ fluctuations between -0.115 and 0.198, could not be readily explained by factors such as land use changes as estimated from the *Hilda+* gridded land cover product (Winkler et al., 2021), seasonality of liquid precipitation input (i.e., rainfall + snowmelt), Parde Coefficients or median rainfall intensity (Fig. R1a,d,g-h). This suggests that some additional drivers, or a combination of drivers, might be influencing the catchment alternating behaviour.

[Figure]

**Figure R1: Land use changes (a-c), Parde Coefficients (d-f), Seasonality of liquid precipitation input (g), and Median rainfall intensity (h) for three example catchments (Kaituna, Thames, and Zschopau) across five 20-year periods ($T_1$–$T_5$).**

In addition, we examined the Thames River in the UK, which also exhibited an alternating sequence of negative $\varepsilon_{IE\omega}$, i.e. reduced evaporation, and positive $\varepsilon_{IE\omega}$, i.e. increased evaporation (Fig. R2). These fluctuations were found qualitatively consistent with land use changes from *Hilda+* data (Fig. R1b). From $T_1$ to $T_2$, a ~5 % decrease in forest cover likely contributed to negative $\varepsilon_{IE\omega}$. Between $T_2$ and $T_3$, the positive $\varepsilon_{IE\omega}$ correlates with a 4.3 % increase in pasture and a 1.7 % increase in grass/shrubland, with negligible change in forest cover. The subsequent decrease in evaporation from $T_3$ to $T_4$ coincides with a 1 % reduction in forest cover and a ~2.5 % decrease in grass/shrubland. However, during the final period ($T_4$ to $T_5$), vegetation changes cannot explain the observed alternating behaviour.

[Figure]

Figure R2: Individual distribution of deviations ($\varepsilon_{IE\Delta1}$, $\varepsilon_{IE\Delta2}$, $\varepsilon_{IE\Delta3}$ and $\varepsilon_{IE\Delta4}$) for Thames River in the UK.

The shifting behaviour of $\varepsilon_{IE\omega}$ for the Zschopau River (Fig. 6n) into one dominant direction after $\varepsilon_{IE\Delta1}$ coincides with a gradual decrease in the seasonality of liquid precipitation input (i.e. rainfall + snowmelt; Fig. R1g), combined with an increase in forest cover (*Hilda+* data) towards the end of century (Fig. R1c). Renner et al. (2014) and Renner and Hauffe (2024) also reported a gradual forest recovery in the Zschopau catchment during this period, which could further contribute to the observed shift.

While some of these explorations suggests potentially plausible correlations with land use changes and seasonality of liquid precipitation, the available data is insufficient to draw robust conclusions.

We have incorporated these findings in the revised manuscript as follows:

p. 15, lines 421-424 (Results):

*"In contrast, more variable patterns were found for other catchments (Fig. 6g-o). For example, in the Sava River at Radece (Slovenia; 6004 km²; ID SI_0000007) the four distributions of the annual deviations all display a wider spread, with IQR ~ 0.113, indicating a higher degree of storage fluctuation between individual years (Fig. 6i).* **This variability may largely be attributed to hydropower developments and the associated changes in hydropower production levels (Levi et al., 2015), which disrupt natural flow regimes by increasing runoff during high demand and altering seasonal flow patterns (Renofalt et al., 2010; Lee et al., 2023)***. In addition, the medians do considerably deviate from zero, as indicated by median $\varepsilon_{IE\omega}$ ranging between -0.023 and 0.118 (Fig. 6h)."*

p. 18, lines 494-498 (Results):

*"In contrast, 7 % of the catchments were tagged as "Alternating" and a dependency between $I_{E,i}$ and $\varepsilon_{IE\omega}$ could not be ruled out. A characteristic example for this type of catchments is the Kaituna*

catchment (New Zealand;  706 km², ID NZ_0000003) in Fig. 6j-l. This catchment features major fluctuations with median $\varepsilon_{IE\omega}$ between -0.115 and 0.198. In addition, although no systematic evolution of median $\varepsilon_{IE\omega}$ over time was evident (Fig. S4d), the data suggest the potential presence of a dependency on $I_{E,i}$ as shown in Fig. S4c. **The pronounced alternating behaviour of the $\varepsilon_{IE\omega}$ fluctuations between -0.115 and 0.198, could not be readily explained by factors such as land use changes as estimated from the Hilda+ gridded land cover product (Winkler et al., 2021), seasonality of liquid precipitation input (i.e., rainfall + snowmelt), Parde Coefficients or median rainfall intensity (Fig. S3a,c,e-f). This suggests that other additional drivers, or a combination of drivers, influence this catchment's alternating behaviour.**"

p. 18, lines 502-505 (Results):

"The remaining 102 catchments (4 %) were tagged as "Shift", as they exhibit a rather consistent shift of median $\varepsilon_{IE\omega}$ over time. This can be seen for a selected example in Fig. 6m-o.The median $\varepsilon_{IE\omega}$ in this catchment of the Zschopau River (Germany; 1544 km2; ID DE_0000027) does not only significantly vary between -0.055 and 0.037 but it does so rather systematically into one dominant direction after $\varepsilon_{IE\Delta1}$ ("+ - ++"; Fig. 6n). **This shift coincides with a gradual decrease in the seasonality of liquid precipitation input (i.e., rainfall + snowmelt; Fig. S3e) and an increase in forest cover towards the end of century as estimated from Hilda+ data (Fig. S3b). Additionally, Renner et al. (2014) and Renner and Hauffe (2024) reported a gradual recovery of forests in the Zschopau catchment during this period, which may further contribute to the observed shift.**"

*Reviewer comment:*

*Title: The title says that most catchments deviate, but the conclusion states "62 % do not significantly deviate", which is contradicting.*

Reply:

Please note that we have made the deliberate choice to formulate the title as "Catchments do not strictly follow….". This is more general than "Most catchments do not strictly follow…" and does not directly imply a majority.

Indeed, our analysis suggests that, based on Wilcoxon Signed Rank Tests, for 62 % of the catchments the median deviations were not significantly different from zero. However, it is important to note that minor deviations were still observed in most catchments even though they were not classified as significant based on this specific statistical test. It would probably too naive to assume that $\varepsilon_{IE\omega} = 0$ and that catchments therefore strictly follow their curves, as also demonstrated, for example, by (Reaver et al., 2022). We therefore prefer to reflect this in the title of the paper. We believe this title conveys the key message of this research work. However, we have clarified this point in our revised manuscript.

p. 14, lines. 387-390 (Results):

"Conversely, this also entails that for a majority of 58–66 % of the distributions there is less evidence (i.e., p > 0.05) that the median $\varepsilon_{IE\omega}$ are different form zero. **Note that minor $\varepsilon_{IE\omega}$ were observed in most catchments. Although these $\varepsilon_{IE\omega}$ were not classified as significant based on the Wilcoxon Signed Rank Test used here, it may be too naive to assume that the deviations $\varepsilon_{IE\omega}$ are strictly zero, as also demonstrated by Reaver et al. (2022).** Overall, this is consistent with results from previous studies and shapes a picture in which catchments do not strictly and necessarily follow their expected parametric $I_E$ curves, but that the deviations thereof remain close to zero or very limited for many catchments."

p. 27, lines. 760-761 (Conclusions):

*"62 % of the catchments do not significantly deviate from their expected parametric Budyko curves, **although minor deviations were still observed.** However, this also entails that a fraction of 38 % does indeed deviate."*

*Reviewer comment:*

*l. 69  Climate is not the only driver; do not forget changes in water and land use, which have been broadly found to drive the trajectory of movement in Budyko space.*

Reply:

We completely agree. We have expanded the statement on p. 3, lines 71-75 (Introduction) as follows:

*"Recently, it was also argued that catchments should not be necessarily expected to follow their long-term average, catchment specific parametric Budyko curves when subject to climatic perturbations, expressed as changes in $I_A$ (Berghuijs and Woods, 2016; Reaver et al., 2022; Jaramillo et al., 2022; Jaramillo et al., 2018). Such deviations ($\varepsilon_{IE\omega}$) from the expected parametric Budyko curve, were previously referred to as residual or landscape-driven, **indicating that many factors other than $I_A$, such as human-induced changes in water and land use (e.g. afforestation, deforestation, irrigation, reservoir construction) also play a  role (Donohue et al., 2007; Wang and Hejazi, 2011; Destouni et al., 2013; Sterling et al., 2012; Van Der Velde et al., 2014; Jaramillo and Destouni, 2015; Levi et al., 2015; Nijzink et al., 2016; Daly et al., 2019; Hrachowitz et al., 2021; Gan et al., 2021)**."*

*Reviewer comment:*

*Fig. 2- The use of symbols in variables became too confusing at some point. The subindices in the variables are long and have a long set of characters, as shown in Fig. 2. Maybe this can be simplified in some way. In the same way, the critical variable $\varepsilon_{IE\omega}$ is not explicitly shown in Fig. 2. There are also some inconsistencies, e.g., $\varepsilon_{IE\Delta j}$. Why "j"? I would avoid the i+1 subindex in the figure so that it agrees with the variables called in the text.*

Reply:

We agree with this observation. Given the multi-decadal periods involved, simplifying the notations are quite challenging without losing clarity. We have explicitly shown $\varepsilon_{IE\omega}$ in the revised figure and addressed any inconsistencies throughout the revised manuscript.

*Reviewer comment:*

*Fig. 5.  $\varepsilon_{IE\Delta}$ does not agree with its expression in Fig. 2. This also brings confusion. Please double-check these issues across the manuscript.*

Reply:

We thank the reviewer for pointing this out. We have corrected it and ensured consistency in the use of symbols throughout the revised manuscript.

*Reviewer comment:*

*L. 186 Why are the time periods consecutive? I see no problem in comparing the changes from, for example, $T_1$ to $T_4$. These new permutations would give even more robustness to the statements of deviation or not deviation.*

Reply:

We do, in principle, agree and considered this approach in the initial phase of the research. However, we chose to proceed with comparing consecutive periods for the following reasons:

1. The use of the most recent 20-year period as the baseline allows to explore the temporal stability of changes. Meaning, is there a systematic pattern over time, as reflected in the classes "Alternating" and "Shift" or can the fluctuations be assumed to be random, as reflected in the classes "Stable" and "Variable"? Using permutations of different periods and thus not preserving the temporal sequence would make such a distinction problematic. We believe that such a classification is necessary for the interpretation of any type of future estimates of $I_E$. For catchments in the classes "Alternating" and "Shift", historical data indicate a change of the underlying distributions which need to be considered for any type of future estimation. In the absence of further information, it is then plausible to assume that the most recent distributions are more representative as baseline for predictions.
2. In many cases, we do not have complete data for five 20-year periods across the full 100-year timeframe. As only 159 out of 2,387 catchments include the full 100-year period for $\Delta_{1-2}$, while 889 catchments have data for $\Delta_{2-3}$, and the number increase to 1916 for $\Delta_{3-4}$, and 2269 for $\Delta_{4-5}$. This distribution shows that for most catchments, only two or three 20-year periods are available, making changes to the baseline period less impactful.

However, for completeness, we have conducted an analysis in which we fixed the oldest available 20-year period as the baseline and calculated distributions of deviations for the subsequent 20-year periods accordingly. The results of this analysis are discussed in the Supplement (Fig. S6a-f). Furthermore, we have added this point to the methods and results section of the revised manuscript.

 p. 8, lines. 201-204 (Methods):

*"For each catchment we then used $\omega_i$ from each time period $T_i$ to compute the expected $I_{E,i+1}$ for the subsequent period $T_{i+1}$ (i.e. point $B_{Ti+1}$\* in Fig. 2). This then allowed to estimate the individual deviations of the 20 annual observed $I_{E,o}$ values from the expected $I_{E,i+1}$ curve. For each pair of time periods $T_i$–$T_{i+1}$ (i.e. $T_1$–$T_2$, $T_2$–$T_3$, etc., hereafter referred to as $\Delta_{1-2}$, $\Delta_{2-3}$, etc.) this resulted in an individual distribution of annual deviations $\varepsilon_{IE\Delta j}$ around a 20-year average in each catchment (Fig. 3b). **This approach using a temporally changing (dynamic) baseline was chosen as it is more sensitive to capture trends and shifts in hydrological behaviour of catchments over time than a fixed baseline. For completeness, we also performed the same analysis by using a fixed baseline (i.e., using the earliest available period as a fixed baseline) and provide the results thereof in the Supplement."***

p. 18-19, lines. 523-525 (Results):

*"In contrast, while also featuring a marginal distribution with a median deviation $\varepsilon_{IE\omega}$~ 0.021, the Sava River catchment (Fig. 6i), tagged as "Variable", is characterized by a considerably wider scatter of the annual deviations around the median, as evident by the higher IQR of ~0.113. Three additional illustrative examples of well-known river basins are presented in Fig. S5. **In contrast, the analysis,***

*which uses the earliest available period as a fixed base line, shows an increase in the number of "Stable" catchments along with a slightly higher median $\varepsilon_{IE\omega}$ values. Further details are provided in the Supplement (Fig. S6a-f)."*

*Reviewer comment:*

*L. 200 ω is both the Budyko and PDF scaling parameters, which is also confusing.*

Reply:

We thank the reviewer for this observation. In the revised manuscript, we have replaced the scale parameter ω of Skew Normal Distribution with λ to avoid confusion.

*Reviewer comment:*

*Fig. 3 Mention the example of the basin you are showing here.*

Reply:

This figure is not based on real data and is intended for illustration purposes only. We have clarified this in the caption of Figure 3 in the revised manuscript.

p. 9, lines. 256-257 (Methods):

*"Figure 3:  Flow chart of methodology. Step 1: Estimation of catchment-specific $I_{E,i}$ curves and the distribution of annual $I_{E,o}$ around it for each period $T_i$. Step 2: Distributions of annual deviations $\varepsilon_{IE\Delta j}$ from expected $I_{E,i+1}$ between subsequent time periods Step 3: Fit parametric distributions to the empirical distributions of annual deviations $\varepsilon_{IE\Delta j}$. Step 4: Evaluate temporal stability of the distributions $\varepsilon_{IE\Delta j}$ in subsequent pairs of time periods Step 5:  Aggregated long-term marginal distribution of annual deviations $\varepsilon_{IE\omega}$ from expected $I_E$ for each catchment. Step 6: Evaluation of the sensitivity of the marginal distributions of annual deviations $\varepsilon_{IE\omega}$ to the choice of 20-year averaging window. **Note, the generated distributions of $\epsilon_{IE\omega}$ are illustrative examples that are not based on real data."***

*Reviewer comment:*

*Fig. 6. Can you classify the catchments with the classification of Table 2? This helps understand which ones correspond to which. Also, why did you choose these catchments? I would also put the name of the catchment in the plots.*

Reply:

Excellent suggestion. In the revised manuscript, we have mentioned both the class and names of each catchment in the plots. Furthermore, the selected example catchments were chosen because they provide a good representation of the different categories used in this research.

*Reviewer comment:*

*L. 395 The answer to your finding about the Sava River may be found in Levy et al. (2015); hydropower development.*

Reply:

We thank the reviewer for pointing us to that paper. We have included relevant information in the results section of the revised manuscript.

p. 15, lines 421-424 (Results):

*"In contrast, more variable patterns were found for other catchments (Fig. 6g-o). For example, in the Sava River at Radece (Slovenia; 6004 km$^2$; ID SI_0000007) the four distributions of the annual deviations all display a wider spread, with IQR ~ 0.113, indicating a higher degree of storage fluctuation between individual years (Fig. 6i).* **This variability may largely be attributed to hydropower developments and the associated changes in hydropower production levels (Levi et al., 2015), which disrupt natural flow regimes by increasing runoff during high demand and altering seasonal flow patterns (Renofalt et al., 2010; Lee et al., 2023)***. In addition, the medians do considerably deviate from zero, as indicated by median $\varepsilon_{IE\omega}$ ranging between -0.023 and 0.118 (Fig. 6h)."*

*Reviewer comment:*

*Fig. 8 Any use for the palette change in Fig. 8a?*

Reply:

The palette in Fig. 8a differs from that in Fig. 8b to match the palette used in Fig. 7b. This was done to maintain consistency and help the reader to better understand the figures. In contrast, Fig. 8b uses a single colour as the interquartile range (IQR) values are all positive, and thus a more varied colour palette was not necessary. The choice of colour is just random. We have clarified the use of the different colour schemes in the figure caption of the revised manuscript.

p. 19, lines 527-529 (Results):

*"Figure 8: Visualization of long-term a) Median $\epsilon_{IE\omega}$ and b) Interquartile Range (IQR) of $\epsilon_{IE\omega}$ for aggregated long-term marginal distribution of $\epsilon_{IE\omega}$ across all catchments (2387) along with the corresponding Cumulative Distribution Function (CDF).* **The varying colour palette in Fig. 8a aligns with the palette used in Fig. 7b to maintain consistency. In Fig. 8b, a uniform colour is used since IQR values are all positive."**

*Reviewer comment:*

*Discussion: I would like to know the thoughts from the authors on the future use of the framework for identifying human modifications to the water cycle, as it has largely been used to date. Maybe some recommendations on the way forward for this goal could be included in the discussion.*

*For instance, the fact that most basins do not deviate does not necessarily mean that the Budyko framework (and the authors' approach) cannot be used to continue identifying human drivers of change. In fact, such identification relies on the deviations to recognize drivers of change. A way forward can be the categorization of the authors into stable, variable, alternating, and shifting categories and to focus analysis on some of these groups.*

Reply:

We completely agree that despite the minor deviations observed in the majority of catchments, the Budyko framework remains useful for identifying human-driven changes. Indeed, deviations are key for recognizing these drivers of change. Categorizing catchments into "Stable", "Variable", "Alternating" and "Shift" could guide a targeted future research. For example, catchments in the "Alternating" and "Shift" categories may in the past either have been subject to more substantial human interference than those in the other categories or they may be more *sensitive* to human-induced changes. Further investigations into the drivers of these deviations could enhance our understanding of how these human-induced changes influence catchments responses differently in different environments. We have incorporated and discussed this perspective in the revised manuscript by adding following lines:

p-26, lines. 736-743 (Discussion):

*"Despite the challenges associated with catchments classified as "Alternating" and "Shift" the Budyko framework remains useful for identifying human-driven changes to the water cycle. Although many catchments showed only minor deviations, these deviations are key for recognizing drivers of change. Categorizing catchments into "Stable", "Variable", "Alternating" and "Shift" can guide targeted future research. For example, catchments in the "Alternating" and "Shift" categories may in the past either have been subject to more substantial human interference than those in the other categories or they may be more sensitive to human-induced changes. Further investigations into the drivers of these deviations may strengthen our understanding of how human-induced changes influence catchments responses differently in different environments."*

*Reviewer comment:*

*Could the authors provide a list in Supplementary on the catchments that fall in each of the categories (if this is not already mentioned).*

Reply:

We thank the reviewer for this suggestion. A detailed summary table, including the list of catchments with their respective categories, is available in a separate repository available at: https://doi.org/10.5281/zenodo.10925966. We have added this to our data availability statement.

p.27, lines. 774-780

**References**

[revised manuscript text omitted]

**Reviewer 2**

We would like to thank the referee for the detailed comments. We greatly appreciate the time and effort taken in thoroughly reviewing our manuscript and for providing valuable and insightful perspectives on our research. We will carefully consider these comments while revising the manuscript.

We have separated the different comments (*shown in italic*) and provide our replies below. Text in the original manuscript is shown in '*italic*' and revised text in '***bold***'.

*Reviewer comment:*

*After a thorough read of the article "Catchments do not strictly follow Budyko curves over multiple decades but deviations are minor and predictable" by Ibrahim et al., I can see the amount of work and understand the main arguments of the authors. The goal is to assess the predictive power of the parametric Budyko curves, usually considered as not suitable for climate projections since they rely on a semi-empirical parameter, and the lack of physical explanation behind it questions whether fixing it to project future behaviours of catchments is pertinent. The authors show that over most of the catchments studied, from one 20-year period to the next, the distribution of deviations to the predictive curve is minimum and stable. This leads them to conclude that the Budyko framework can be used for projections under a changing climate, just considering a stable distribution of deviation around the curve as a shape of uncertainty.*

Reply:

We would like to thank the reviewer for a thorough review and for highlighting the objective of our research. We appreciate the acknowledgement of our efforts to evaluate the predictive power of the parametric Budyko curves.

*Reviewer comment:*

*The article is well written, well-illustrated and well-integrated into the current literature. However, I am not sure every steps of the method are pertinent and I am not fully convinced by the conclusions drawn and how new the results are.*

Reply:

We highly appreciate the reviewer's encouraging feedback on the writing, illustration and integration of our manuscript into the current literature.

*Reviewer comment:*

*The method compares successive periods of 20 years. The method stays pertinent when looking at a 20-year period and looking whether or not the median deviation from the curve can be considered different from zero or not (step 2). Therefore, the conclusions can only be applied to argue that the Budyko framework can be used for 20-years projections, which is rarely the temporality used for climate projections.*

We acknowledge the observation that the chosen approach is valid strictly for predictions over 20-year windows, while less so for longer-range predictions. In the early phase of the study design, we have alternatively also considered longer time windows, but eventually, deliberately decided to use windows of 20 successive years as an approach that balances the need for sufficiently long time periods to limit the effect of storage changes d$S$/d$t$ (Han et al., 2020), while preserving the temporal sequence in the data that allowed us to place each catchment into a specific category (i.e. "Stable", "Variable", "Alternating" or "Shift"). This aspect is one of the major novelties of our analysis, as it has – to our knowledge – never been analysed on global scale before. The use of fewer but longer time windows, such as ~ 30 years, as previously done by others, e.g., Destouni et al. (2013), would have considerably limited a meaningful distinction of systematic shifts from more random fluctuations over time.

We would also like to emphasize that 20-year periods are a not uncommon time-horizon for many water resources management interventions and planning, where such shorter-term predictions are often more relevant for decision-making.

However, we completely acknowledge the limitations of our choice. We have therefore added our reasoning for the 20-years window and a discussion of the implications of this choice in the revised manuscript as follows:

p. 7, lines. 161-167 (Methods):

*"Here, we have sub-divided the available data records of each catchment into up to five individual 20-year periods Ti over the last century (Table 1).* **This 20-year period was chosen deliberately to balance the need for a sufficiently long period to minimize the impact of storage changes, while preserving the temporal sequence in the data that allowed us to place each catchment into a specific temporal stability category (as described in Step-4). We assume that 20-year periods are long enough to satisfy dS/dt≈0, supported by Han et al. (2020) , who demonstrated that in more than 80 % of catchments worldwide, dS/dt is less than 5 % over 20-year periods. Using longer periods, such as 30 years as used in previous studies (e.g. Destouni et al. (2013)), would have smoothed out potential shifts and limited the ability to detect systematic changes. In addition, 20-year periods align with planning horizons in many water resource management decisions."***

p. 24, lines. 690-691 (Discussion):

*"This further allows some confidence to plausibly assume that $ε_{IEω}$ and the associated $I_E$ under projected future hydro-climatic conditions can, at least for several decades, be robustly predicted based on these distributions.* **However, it is important to note that the 20-year time periods used in this study, while effective for medium-term projections, may limit the ability to make long-term climate projections."***

*Reviewer comment:*

*The method also compares successive deviation distribution, for instance to define "stable" catchments as catchments for which the deviation to the curve from one 20-year period to the next has no specific direction. However, if I understood correctly, each distribution of deviation to the curve for each 20-year period is calculated around a different curve (with the actualised parameter fitted over the previous 20-year period). Then, what if there is a trend in this parameter? I understand it is not possible to evaluate such a trend significantly due to the length of the data but it would invalidate the*

*comparison of the successive distributions. Why not use the same curve for all periods and look if the distribution around the curve changes over time? Could the successive fit over the 20-year sliding time periods be used here to assess trends?*

Reply:

We greatly appreciate the reviewer's sharp observation and agree that a fixed baseline can provide additional insights. To explore this, we conducted the analysis using the reviewer's proposed approach by fixing the first 20-year period as the baseline and calculated distributions of deviations for the subsequent 20-year periods accordingly.

This comparison is illustrated (Figs. R1 and R2) below, in which we compare the original approach with a dynamic baseline to the use of a fixed baseline for two of the example catchments of our study. For the first example catchment (Chemung River, Fig. 6a-c in the manuscript), we observed that the results from both methods were almost identical (Fig. R1a-b & R2a-b). However, for the second example catchment (Lee River Fig. 6d-f in the manuscript), the median deviations were somewhat higher when using a fixed baseline (Fig. R1c-d & R2c-d).

We also extended this analysis to all other study catchments. However, please note that we could include the full 100-year period in only 159 out of 2387 catchments. For the other catchments, we used the oldest available 20-year period as the fixed baseline. We have found that the proportion of "Stable" catchments increased from 72 % (dynamic baseline - original approach) to 84 % (fixed baseline - proposed approach) (Fig. R3), suggesting that dynamic baselines are more sensitive for the detection of systematic changes, i.e. "Shift" (trends) or "Alternating" behaviour. In contrast, while the number of "Stable" catchments increased, we also observed that the median deviations of the aggregated marginal distributions of $\varepsilon_{IE\omega}$ were slightly higher when using a fixed baseline (Fig. R4).

Despite the fact that there are some interesting results, we still prefer to keep the temporally changing (dynamic) baseline in the main part of our analysis for the following reasons:

1. The use of the most recent period as the baseline for assessing temporal stability of catchments is particularly relevant for future predictions, as the most recent data are most likely to provide a meaningful representation of current conditions. Using the oldest period as the baseline may not reflect recent conditions and could result in misleading conclusions, in particular when the first and last periods are far apart.
2. As previously mentioned, only 159 out of 2,387 catchments include the full 100-year period for $\Delta_{1-2}$, while 889 catchments have data for $\Delta_{2-3}$, and the number increase to 1916 for $\Delta_{3-4}$, and 2269 for $\Delta_{4-5}$. This distribution shows that for most catchments, only two or three 20-year periods are available, making changes to the baseline period less impactful.
3. As the reviewer correctly pointed out, it is difficult to quantify trends in the omega parameter based on only 5 values due to the limited length of the available data records. In many cases, we are working with just 2 or 3 time periods, making it impossible to detect significant trends.

Overall, adjusting the baseline over time is more sensitive to capturing recent shifts and trends in hydrological behaviour, which helps to assign catchments to one of the four categories.

The reviewer's suggestion to use successive fits over sliding 20-year periods to assess trends is indeed interesting. However, this would introduce dependencies between the periods, as each successive period overlaps with the previous one. This, in turn, would compromise the independence of the data from the individual time periods and potentially bias the results.

Based on the above and to provide the reader with a more comprehensive view of our results, we have, as the reviewer suggested, included the results of using a single fixed baseline (Fig. R1-R4) in the Supplement of the revised manuscript. In addition, we have added the following statements in the revised manuscript:

p. 8, lines. 201-204 (Methods):

*"For each catchment we then used $\omega_i$ from each time period $T_i$ to compute the expected $I_{E,i+1}$ for the subsequent period $T_{i+1}$ (i.e. point $B_{Ti+1}$\* in Fig. 2). This then allowed to estimate the individual deviations of the 20 annual observed $I_{E,o}$ values from the expected $I_{E,i+1}$ curve. For each pair of time periods $T_i$–$T_{i+1}$ (i.e. $T_1$–$T_2$, $T_2$–$T_3$, etc., hereafter referred to as $\Delta_{1-2}$, $\Delta_{2-3}$, etc.) this resulted in an individual distribution of annual deviations $\varepsilon_{IE\Delta j}$ around a 20-year average in each catchment (Fig. 3b). **This approach using a temporally changing (dynamic) baseline was chosen as it is more sensitive to capture trends and shifts in hydrological behaviour of catchments over time than a fixed baseline. For completeness, we also performed the same analysis by using a fixed baseline (i.e., using the earliest available period as a fixed baseline) and provide the results thereof in the Supplement."*

p. 18-19, lines. 523-525 (Results):

*"In contrast, while also featuring a marginal distribution with a median deviation $\varepsilon_{IE\omega}$~ 0.021, the Sava River catchment (Fig. 6i), tagged as "Variable", is characterized by a considerably wider scatter of the annual deviations around the median, as evident by the higher IQR of ~0.113. Three additional illustrative examples of well-known river basins are presented in Fig. S5. **In contrast, the analysis, which uses the earliest available period as a fixed base line, shows an increase in the number of "Stable" catchments along with a slightly higher median $\varepsilon_{IE\omega}$ values. Further details are provided in the Supplement (Fig. S6a-f)."*

[Figure]

**Figure R1: Comparison of individual distribution of deviations ($\varepsilon_{IE\Delta 1}$, $\varepsilon_{IE\Delta 2}$, $\varepsilon_{IE\Delta 3}$ and $\varepsilon_{IE\Delta 4}$) between the dynamic baseline (left) and fixed baseline (right) approaches for two example catchments (Chemung River and Lee River) in the "Stable" category.**

[Figure]

**Figure R2: Comparison of long-term marginal distribution of annual deviations between the dynamic baseline (left) and fixed baseline (right) approaches for two example catchments (Chemung River and Lee River) in the "Stable" category.**

[Figure]

**Figure R3: Comparison of temporal stability of the studied catchments using the dynamic baseline (top) and fixed baseline (bottom) approaches. Catchments highlighted with a black border represent the 5 selected examples from Fig. 6 (of the original manuscript), while those outlined in red denote three additional selected example catchments shown in the Supplement (Fig. S5 in the revised manuscript).**

[Figure]

[Figure]

**Figure R4: Comparison of long-term median $\epsilon_{IE\omega}$ values of aggregated marginal distribution of $\epsilon_{IE\omega}$ across all catchments (2387) comparing the dynamic baseline (left) and fixed baseline (right) approaches.**

*Reviewer comment:*

*The authors argue that the distribution around the curve is just a natural variation around the curve ("stable" catchments) or due to regular climatic cycles ("variable" catchments). However, not all catchments fit in these categories, and since there seems to be no homogeneity in the spatial distribution or climatic characteristics of these catchments, it undermines the conclusion that the framework can be used for prediction in most catchments. It is not a generality, since such a study would need to be lead first in a catchment to check that it fits in a "stable" distribution, and whether or not it will seems arbitrary.*

Reply:

While we agree with the reviewer's observation that not all catchments fit into the categories of "Stable" or "Variable", it is important to note that these catchments ("Alternating" or "Shift") constitute only a small minority, comprising 11 % of the study catchments. The remaining 89 % of catchments are either "Stable" or "Variable".

As pointed out by the reviewer, there is no clear spatial pattern. The regional distributions of $\epsilon_{IE\omega}$ remain, with medians of $\sim 0 - 0.02$ (Fig.9 in the original manuscript), broadly consistent with the global distribution (Fig.8) but also with each other across most spatial and climatic classes. This indeed suggests that the overall pattern is rather homogenous, and regional/local effects remain limited. The presented distributions (Figs. 8, 9) are in the absence of further information nevertheless useful to quantitatively estimate the uncertainty for any specific catchment based on past information in a probabilistic way, as clearly pointed out by Montanari and Koutsoyiannis (2014): "*[…] If a deterministic description of the process statistics along time, applicable to future times, is not available, which implies that non-stationarity is impossible to define, the only way for making predictions is through assumptions of stationarity*". Whereby "*if a deterministic description […] is not available*" in our cases corresponds to the impossibility to identify "Shift" and "Alternating" out-of-sample catchments and "*assumption of stationarity*" corresponds in our case to the assumption that out-of-sample catchments are largely "Stable" and "Variable".

We have discussed these points in more detail and have revised our conclusions to reflect that while the framework works well for the wide majority of catchments, it cannot take into account systematic shifts or alternating behaviour in out-of-sample catchments.

p. 25-26, lines. 725-735 (Discussion):

*"It is important to note that approximately 89 % of the study catchments are either "Stable" or "Variable" and with only a small minority (~11 %) exhibiting "Alternating" or "Shift" behaviour. This predominance of "Stable" or "Variable" catchments supports the broad applicability of the Budyko framework for predictive purposes. Although there is no clear spatial pattern, the regional distributions of $\epsilon_{IE\omega}$ remain, with medians of ~ 0 – 0.02 (Fig.9a), broadly consistent with the global distribution (Fig.8a) but also with each other across most spatial and climatic classes. This indeed suggests that the overall pattern is rather homogenous, and regional effects remain limited, making probabilistic predictions feasible in the absence of a deterministic description (Montanari and Koutsoyiannis, 2014). Thus, the presented distributions (Figs. 8a,9a) are in the absence of further information useful to quantitatively estimate the uncertainty for any specific catchment based on past information in a probabilistic way. However, caution is advised for out-of-sample catchments, where the assumption of stationarity may lead to less reliable predictions, as the framework cannot take into account systematic shifts or alternating behaviour."*

p. 27, lines. 767-768 (Conclusions):

*"For 89 % of the study catchments, $\varepsilon_{IE\omega}$ can be considered highly or at least moderately well predictable based on historical data, as distributions of $\varepsilon_{IE\omega}$ in the past were shown to be stable over multiple time periods or characterized by variable fluctuations. **The framework works well for most catchments; however, for out-of-sample catchments showing systematic shifts or alternating behaviour, additional analysis may be required.**"*

*Reviewer comment:*

*I feel the results would benefit from a different presentation, to help show their impact. As briefly presented on the discussion, I feel it would be more pertinent to express the deviation to the curve by how much it shifts the predicted aridity or discharge (%), rather than present changes in an abstract parameter. The impact of the shift in the parameter is different depending on the aridity of the catchment, which could be interesting to analyse and could shed the results in a different light.*

Reply:

We thank the reviewer for this excellent suggestion. We agree that expressing the deviation from the curve in terms of the percentage change in the evaporative index or discharge (or runoff coefficient) provides a better understanding of the impact. We have calculated the percentage change in discharge for each catchment and will include this analysis in the revised manuscript (Fig. R5). Additionally, we have analysed these changes across four different aridity classes to further highlight how the impact of the parameter shift varies with catchment aridity (inset in Fig. R5). This approach has allowed for a more meaningful presentation of the results and has been incorporated into the revised manuscript. This figure has been added as Figure 7c in the revised manuscript with following additions:

p. 1, lines. 21-23 (Abstract):

*"On average, it was found that the majority of 62 % of study catchments did not significantly deviate from their expected parametric Budyko curves. From the remaining 38 % of catchments that deviated from their expected curves, the long-term magnitude of median deviations remains minor, with 70 % of catchments falling within the range of ±0.025 of the expected evaporative index. **When these**

*median deviations were expressed as relative changes in discharge, catchments in arid regions showed higher susceptibility to larger discharge shifts compared to those in humid regions."*

p. 19, lines. 534-538 (Results):

*"Overall, it can be observed that median deviations $ε_{IEω}$ close to zero are dominant globally, with no obvious spatial clustering of more pronounced deviations (Fig. 7b). However, it can also be seen that there is some geographic grouping in the direction, i.e. the sign, of the median $ε_{IEω}$. While for many catchments in the central US and southern Brazil median deviations are negative, i.e. $ε_{IEω} < 0$, the rest of the study catchments globally are dominated by $ε_{IEω} > 0$.* **Overall, median deviations resulted in regionally distinct relative changes in Q across the studied catchments with around ~68 % of the catchments exhibiting changes ΔQ of less than  ±10 % (Fig. 7c). However, catchments in some regions, notably in central US and Southern Africa, can be characterized by ΔQ exceeding ±25 %. Overall, the results indicate that catchments in more arid regions ($I_A$>2) are particularly susceptible to relative changes in discharge as compared to more humid regions (inset Fig. 7c)."**

p. 26, lines. 751 (Discussion):

*"A complete list of the parameters and robustness flags of the individual 20-year distributions as well as of the local aggregated marginal distributions* **with associated changes in Q** *for each of the 2387 study catchments, but also of the regional distributions as stratified by latitude and $I_A$ are provided in the Supplementary data (https://doi.org/10.5281/zenodo.10925966)."*

p. 27, lines. 763-764 (Conclusions):

*"The overall magnitude of deviations is minor. For ~70 % of the catchments the median deviations do not exceed $ε_{IEω}$ = ± 0.025, which is equivalent to ~ 1–4 %, depending on $I_E$.* **These median $ε_{IEω}$, when expressed as relative changes in Q, result in less than a ±10 % change in discharge for most catchments."**

[Figure]

**Figure R5: Spatial distribution map of changes in Q as a result of long-term median $ε_{IEω}$ values. Histogram and cumulative density of changes in Q and change in Q across different $I_A$ bins are presented as two insets. Catchments highlighted with a black border represent the 5 selected examples from Fig. 6 (of the original manuscript), while those outlined in red denote three additional selected example catchments shown in the Supplement (Fig. S5 of the revised manuscript). The boxes represent the 25th to 75th percentiles, while whiskers extend to the 10th and 90th percentiles. Diamonds denote the arithmetic mean, and outliers are not shown.**

*Reviewer comment:*

*Furthermore, with raw values of the shift in the parameter, it is difficult to understand whether it is a negligible change or not, as argued in the conclusion. Having more understandable orders of magnitude of this shift and the associated uncertainty would help argue that there is a potential in using a parametric equation for projection, with an inevitable associated uncertainty, which could be not be wider than the uncertainty associated to climate projection or to physical-based models. This is however still not a very new argument, and should be made with an understanding of the counter-arguments, that we are never sure that empirical models will respond reasonably when faced with unprecedented changes in climate. I believe this study would be interesting in that regard, as, if it doesn't introduce completely new concepts, it has a broader perspectives and a more targeted objective to quantify the uncertainty associated to the deviation from the parametric curve for a catchment in the Budyko framework. It would benefit from being formulated as such.*

Reply:

We agree that it can be difficult to interpret the relevance of change based only on raw values of the shift in the parameter. The reviewer's suggestion to show the magnitude of the shift in forms of a magnitude change such as % change in discharge is indeed very helpful and we have included this in the revised manuscript. Furthermore, we also fully agree that it is an inherent weakness of empirical models that they typically cannot deal with previously unseen changes of the underlying distribution of a specific variable. Identifying the sensitivity of catchments to such changes in underlying distributions to address exactly this issue was our main intention with the classification of the study catchments into four categories. Following both, the shorter term climatic variability but also the long-term changing climate over the past century, catchments classified as "Alternating" or "Shift", are more likely to have experienced changes in the underlying distributions and are thus plausible to remain sensitive to change in the future. For such catchments, the empirical model will thus be less robust for predictions. However, for the other categories i.e., "Stable" or "Variable", the catchments exhibited much less sensitivity to past climatic variability. In the absence of statistical evidence for changing distributions it is thus not implausible to assume that they remain, at least for the near-future, rather insensitive to change and that the empirical models can thus provide plausible predictions.  However, we explicitly acknowledge that issue remains a limiting characteristic of statistical models. We have included these points in the discussion section of the revised manuscript as follows:

p. 25, lines. 717-724 (Discussion):

**"Additionally, empirical models like the Budyko framework have inherent weakness in dealing with previously unseen changes of the underlying distribution of a specific variable (here: $\varepsilon_{IE\omega}$). Our classification of catchments into "Stable", "Variable", "Alternating" and "Shift" categories aims to capture varying levels of sensitivity to changes in underlying distributions. Catchments classified as "Alternating" or "Shift", are more likely to have experienced large changes in the underlying distributions and may thus remain sensitive to future changes, making empirical model less robust for predictions in these cases. Conversely, "Stable" and "Variable" catchments exhibited much less sensitivity to past climatic variability. In the absence of statistical evidence for changing distributions, it is reasonable to assume that they remain relatively insensitive to change in the near future, allowing empirical models to provide plausible predictions."**

*Reviewer comment:*

*Abstract, l11: I think "behaviour" is not the right term. You consider in your study parametric curves, where the parameter is generally considered to represent the specific behaviour of a catchment. A move along the curve is supposed to represent the changes in the catchment responses under a changing climate but with a fixed behaviour.*

Line 11: "*A movement along a Budyko curve with changes in the climatic conditions has been used as a predictor for catchment behaviour under change*"

Reply:

Indeed. We have modified line 11 as follows:

"**A common assumption is that movement along a specific Budyko curve with changes in the aridity index over time can be used as a predictor for catchment responses to changing climatic conditions.**"

*Reviewer comment:*

*L176-179: Here your two sentences are contradictory. If I understood correctly, for each 20-year sub-period, you fitted the curve to the set of n=20 values, not to the 20-year average directly. Therefore, you need to change the first sentence of that paragraph which says the exact opposite.*

L 176-179: *For each catchment and each individual 20-year time period $T_i$, the catchment-specific parametric Budyko curve $I_{E,i}$ defined by parameter $\omega_i$ is obtained by fitting Eq.(2) to the mean annual positions of each catchment in the Budyko space, as computed from the observed water balance data.*

Reply:

We thank the reviewer for pointing this out. We have modified the lines 176-179 (revised manuscript lines 188-190) as follows:

"**For each catchment and each individual 20-year time period $T_i$, the catchment-specific parametric Budyko curve $I_{E,i}$ defined by parameter $\omega_i$ is obtained by fitting Eq.(2) to the set of 20 annual values of each catchment in the Budyko space, as computed from the observed water balance data.**"

*Reviewer comment:*

*I really like Figure 3, it helps understanding the steps of the method.*

Reply:

We are glad that the reviewer found this figure helpful.

*Reviewer comment:*

*Paragraph 3.1: I am not sure I understand the pertinence of that part of the results. Is there a point in comparing the changes in climate variables at the global scale? Would it not be more pertinent to look at these changes in different groups of catchments, for instance looking to see if they relate to the categories of "stable", "variable", "alternating" or "shifting" catchments? Or geographically?*

Reply:

We thank the reviewer for the valuable feedback and suggestion. The rationale for showing periodic changes in climatic variables, at a global scale throughout the study period, is to provide an overall view of how the key variables that determine the Budyko coordinates ($E_A/P$ and $E_P/P$), have changed over time. This global perspective helps to understand the trends and catchments' movement within the Budyko space. However, we also agree with the suggestion to provide a more detailed description based on the catchment categories for this analysis. We have carried out the proposed analysis and have incorporated it in the revised manuscript by replacing Figure 4 with Figure R6 (see here below).

[Figure]

$n\Delta_{1-2} = 159$ , $n\Delta_{2-3} = 889$, $n\Delta_{3-4} = 1916$ , $n\Delta_{4-5} = 2269$

**Figure R6: Temporal stability category-wise mean 20-year changes in hydro-climatic variables for the studied catchments between two consecutive periods. a) Precipitation $P$, b) Temperature $T$, c) Potential evaporation $E_P$, d) Actual evaporation $E_A$, e) Aridity index $I_A$, and f) Evaporative index $I_E$. The boxes represent the 25th to 75th percentiles, while whiskers extend to the 10th and 90th percentiles. Diamonds denote the arithmetic mean, and outliers are not shown**

Furthermore, following statements are added to the revised manuscript:

p. 13, lines. 344 (Results):

*"These combined factors led to slightly more arid conditions in the first half of the 20th century, followed by a considerable reduction of aridity index $I_A$ and thus to a shift towards somewhat more humid conditions towards the end of the century **across all of the temporal stability categories** (Fig. 4e), in which, **on average** 78 % and 75 % of the catchments showed decreases in $I_A$ for $\Delta_{3-4}$ and $\Delta_{4-5}$, respectively."*

p. 23, lines. 636-637 (Results):

*"Our analysis revealed that most study catchments underwent continuous multi-decadal hydro-climatic fluctuations throughout the 20th century (Fig. 4 & Fig. S1). **Notably, these fluctuations were largely consistent across the different temporal stability categories**. Unlike previous studies comparing only two time periods (Jaramillo and Destouni, 2014), here the higher temporal resolution into with up to five 20-year periods, showed that these fluctuations were not one-directional, with the first half of the century trending towards higher aridity and the latter half towards increased humidity, suggesting cyclic behaviour over longer time scales."*

*L535: You make the argument here that "the spread around the regional medians consistently decreases with increasing IA across all latitude bands" and therefore that "catchments in more humid regions across the study 535 domain are subject to more pronounced annual water storage fluctuations". However, as you say yourself, the impact of the shift is different depending on the aridity of the catchment. Here this argument would beneficiate from presenting relative changes in discharge or IE rather than changes in the parameter.*

Reply:

We completely agree that presenting relative changes in discharge or $I_E$ rather than changes in the parameter would provide a clearer understanding. We have conducted the analysis as suggested and have included the results in the revised manuscript by adding Figure R5 as Figure 7c in the revised manuscript with additions in the following lines:

p. 1, lines. 21-23 (Abstract)
p. 19, lines. 534-538 (Results)
p. 27, lines. 763-764 (Conclusions)

*L50: The sentence could be reformulated. Maybe the word "described" is unnecessary.*

Line 50: *"The fact that the long-term water balance exhibits such a relatively consistent behaviour described across a wide spectrum of hydro-climatically and physio-graphically distinct environments has led to the hypothesis that the general shape of Budyko curves emerges for natural systems in a co-evolution of climate, soil water storage and vegetation properties."*

Reply:

We thank the reviewer for the suggestion. We have removed the word "described" from the line 50 (revised manuscript line 51).

**"The fact that the long-term water balance exhibits such a relatively consistent behaviour across a wide spectrum of hydroclimatically and physiographically distinct environments has led to the hypothesis that the general shape of Budyko curves emerges for natural systems in a co-evolution of climate, soil water storage and vegetation properties."**

*L80: This sentence is also a little awkward. Especially the last part. Maybe separate it in two sentences.*

Line 80: *"In other words, some level of deviations from the expected parametric Budyko curves realistically needs to be expected as different time periods will never be characterized by exactly the same environmental conditions and as well the mechanistic processes that control ω are not well understood."*

Reply:

We have modified the line 80 (revised manuscript line 82) as follows:

*"In other words, some level of deviation from the parametric Budyko curves is to be expected, as different time periods will never be characterized by exactly the same environmental conditions. However, the mechanistic processes that control these deviations, and thus ω, are not well understood."*

Line 257: "*To do so, we have followed three steps and for each catchment the up to j = 4 distributions of deviations $\varepsilon_{IE\Delta j}$ from expected $I_{E,i+1}$ between subsequent time periods were compared and analysed for their changes over time (Fig. 3, Sub-steps 4.1-4.3).*"

Reply:

We have split the sentence into two, as the reviewer suggested. The modified sentences (revised manuscript line 271) are as follows:

**"To do so, for each catchment the up to j = 4 distributions of deviations $\varepsilon_{IE\Delta j}$ from expected $I_{E,i+1}$ between subsequent time periods were compared and analysed for their changes over time (Fig. 3, Sub-steps 4.1-4.3). We have followed three sub-steps:"**

Line 325: "*Combined this led to slightly more arid conditions in the first half of the 20$^{th}$ century, followed by considerable reduction of aridity index $I_A$ and thus to a shift towards somewhat more humid conditions towards the end of the century (Fig. 4e), in which 78 % and 75 % of the catchments showed decreases in $I_A$ for $\Delta_{3-4}$ and $\Delta_{4-5}$, respectively.*"

Reply:

We have modified line 325 (revised manuscript line 342) as follows:

"**These combined factors** led to slightly more arid conditions in the first half of the 20$^{th}$ century, followed by a considerable reduction of aridity index $I_A$ and thus to a shift towards somewhat more humid conditions towards the end of the century across all of the temporal stability categories (Fig. 4e), in which, on average 78 % and 75 % of the catchments showed decreases in $I_A$ for $\Delta_{3-4}$ and $\Delta_{4-5}$, respectively.*"*

Line 329-330: "*Furthermore, we observed various hydroclimatic conditions over time periods along with indications that there are deviations $\varepsilon_{IE\omega}$ from the expected $I_E$, as elaborated in detail in Fig. S1.*"

Reply:

We have modified these lines (revised manuscript lines 346-350) as follows:

*"The overall movement of catchments in the Budyko space due to hydro-climatic changes are illustrated in Fig. S1 (Jaramillo and Destouni, 2014). If these movements were driven only by changes in $I_A$, catchments would be expected to move within the directional range of $45^o < \alpha < 90^o$ or $225 < \alpha < 270^o$ (Jaramillo et al., 2022). However, observed movement of catchments are also found in other directions, indicating deviations ($\varepsilon_{IE\omega} \neq 0$) from the expected $I_E$, as elaborated in detail in Fig. S1."*

*Reviewer comment:*

*L360: Supplementary material should not be cited before figures from the main article in a given paragraph. Otherwise why not include it?*

Reply:

We have removed the citation of Supplement material in this paragraph and inserted it after elaboration of the previous comment.

---

## Referee Report (RR1)

The revised manuscript "Catchments do not strictly follow Budyko curves over multiple decades but deviations are minor and predictable" is a very improved version of the earlier manuscript. I feel that all the reviewers' comments have been very well addressed, with complete and detailed answers and well-chosen adjustments made accordingly to the manuscript.

I feel the introduction is very complete and interesting; the method is a lot clearer, with the choices made better explained and discussed. The figures are clear, complete and the adjustments made help to better show the richness of the results. I believe that this manuscript is very well integrated into the current literature and the current stakes surrounding models' robustness for predictions. This article would be an interesting contribution to the field, and I recommend it for publication.

Here are still some minor comments and small corrections I would suggest:

I would thank the authors for adding more perspective to the results by showing the effect on discharge changes / IE (%), which are easier to understand than changes in the median. I would maybe specify more clearly that these changes in Q are changes due to the deviation to the curve only (if I understood correctly). There is a predicted change along the Budyko curve, what the authors evaluate is the part that is not predicted due to a deviation to the curve.

Also, how is the median value of deviation interpreted? A unit for the median could be interesting to understand the order of magnitude analysed. Is it a relative variation in IE? Or an absolute variation in IE? If it is the latter, the regional aggregation of median distributions (l 539) would need to be further discussed.

L 78 : missing a parenthesis for the e.g.

L 160: 'Ti', not very clear it is a notation. Maybe be more explicit (noted Ti for the ith period) or refer to Table 1. At minima, write Ti and not Ti, as done later on. Otherwise, it's a very good paragraph.

L 503: "decrease in the seasonality": what does it mean?

---

## Author Response (AR2)

**Note: Adjustments in response to the comments from Reviewer 2 are highlighted in yellow in the revised manuscript.**

**Reviewer 2**

We would like to thank the referee once again for their continued review and thoughtful comments. We greatly appreciate the time and effort taken in further refining our manuscript and for providing additional valuable insights. We have carefully considered these additional comments in the revised manuscript.

We have separated the different comments (*shown in italic*) and provide our replies below. Text in the original manuscript is shown in '*italic*' and revised text in '***bold***'.

*Reviewer comment:*

*The revised manuscript "Catchments do not strictly follow Budyko curves over multiple decades but deviations are minor and predictable" is a very improved version of the earlier manuscript. I feel that all the reviewers' comments have been very well addressed, with complete and detailed answers and well-chosen adjustments made accordingly to the manuscript.*

*I feel the introduction is very complete and interesting; the method is a lot clearer, with the choices made better explained and discussed. The figures are clear, complete and the adjustments made help to better show the richness of the results. I believe that this manuscript is very well integrated into the current literature and the current stakes surrounding models' robustness for predictions. This article would be an interesting contribution to the field, and I recommend it for publication.*

Reply:

We are thankful for the encouraging remarks made by the reviewer on our manuscript. We are pleased that the reviewer found our approach and reflections valuable to the field.

*Reviewer comment:*

*I would thank the authors for adding more perspective to the results by showing the effect on discharge changes / $I_E$ (%), which are easier to understand than changes in the median. I would maybe specify more clearly that these changes in Q are changes due to the deviation to the curve only (if I understood correctly). There is a predicted change along the Budyko curve, what the authors evaluate is the part that is not predicted due to a deviation to the curve.*

Reply:

We highly appreciate the reviewer's thoughtful comment. We have clarified this point in our revised manuscript that the changes in Q presented in our analysis are solely attributable to deviations from the expected parametric Budyko curve. We have modified the caption of Figure 7c as follows:

p. 17, lines 485-487 (Results):

*"Figure 7: a) Temporal stability, b) long-term median $\varepsilon_{IE\omega}$ values map of aggregated long-term marginal distributions for the study catchments, and c) change in Q as a result of long-term median $\epsilon_{IE\omega}$ values. Histogram and cumulative density of changes in Q, and change in Q across different $I_A$ bins are presented as two insets.* ***Change in Q reflect the change due to median deviations $\varepsilon_{IE\omega}$ from the***

*expected parametric Budyko curve only (i.e., excluding any change resulting from a change in aridity and its associated movement along the expected curve).Catchments highlighted with a black border represent the 5 selected examples from Fig. 6, while those outlined in red denote three additional selected example catchments shown in the Supplement (Fig. S5). The boxes represent the 25th to 75th percentiles, while whiskers extend to the 10th and 90th percentiles. Diamonds denote the arithmetic mean, and outliers are not shown."*

*Reviewer comment:*

*Also, how is the median value of deviation interpreted? A unit for the median could be interesting to understand the order of magnitude analysed. Is it a relative variation in $I_E$? Or an absolute variation in $I_E$? If it is the latter, the regional aggregation of median distributions (l 539) would need to be further discussed.*

Reply:

We thank the reviewer for this insightful comment. The deviation ($\varepsilon_{IE\omega}$) is dimensionless because it represents the absolute difference between observed evaporative index and predicted evaporative index (Eq. (4)). Concerning the regional marginal distribution of deviations, these are not based on aggregation of median deviations but rather on aggregating yearly deviations for each catchment within a defined latitude-aridity index bins. To clarify this, the following modifications have been made in the revised manuscript as follows:

p. 3, lines 75-76 (Introduction):

*"Where $\varepsilon_{IE\omega}$ is defined as the **absolute** difference between the observed evaporative index ($I_{E,o}$) and the predicted evaporative index ($I_E$) derived from the expected parametric Budyko curve, **making it dimensionless**."*

p. 6, line 143 (Methods):

*"Figure 2: A schematic representation of a catchment movement in Budyko space between two long-term time periods $T_i$ and $T_{i+1}$. Case A: Catchment A moves along the same Budyko curve from the first period $T_i$ to the next period $T_{i+1}$ (i.e., $\omega_i = \omega_{i+1}$). Case B: Catchment B has deviated from its expected parametric Budyko curve (i.e., $\omega_i \neq \omega_{i+1}$), **resulting in deviation $\varepsilon_{IE\omega,i+1}$ (Eq.(4))**"*

p. 19-20, lines 545-548 (Results):

*"For a more regional evaluation, **the yearly $\varepsilon_{IE\omega}$ values for individual catchments were aggregated into regional marginal distributions of $\varepsilon_{IE\omega}$ stratified according to the long-term mean aridity index $I_A$ and varied latitude bands (Fig. 9a). These regional distributions capture the variability of yearly $\varepsilon_{IE\omega}$ across regions with the median $\varepsilon_{IE\omega}$ serving as a robust measure of central tendency**. The general pattern found across most regions with available data are broadly consistent. 16 out of 20 regions are characterized by median deviations $\varepsilon_{IE\omega}$ that do not exceed ± 0.02. Similarly, no consistent directional pattern in the magnitude of regional median $\varepsilon_{IE\omega}$ could be identified either (Fig. 9b). For higher latitude regions beyond ±30°, the minor fluctuations in median $\varepsilon_{IE\omega}$ bear no evidence for a relationship with $I_A$. On the other hand, the data suggest that the spread around the regional medians consistently decreases with increasing $I_A$ across all latitude bands except 50° N–90° N band as shown by the sequence of IQR in Fig. 9c. This indicates that catchments in more humid regions across the study domain are subject to more pronounced annual water storage fluctuations."*

*Reviewer comment:*

*L 78 : missing a parenthesis for the e.g.*

Reply:

We thank the reviewer for pointing this out. We have added a parenthesis at the end of this sentence in revised manuscript.

*Reviewer comment:*

*L 160: 'Ti', not very clear it is a notation. Maybe be more explicit (noted Ti for the ith period) or refer to Table 1. At minima, write $T_i$ and not Ti, as done later on. Otherwise, it's a very good paragraph.*

Reply:

We thank the reviewer for highlighting this point. In the revised manuscript, we have updated both the text and title of Table 1 to explicitly define $T_i$ as the $i^{th}$ 20-year period and referred to Table 1 for clarification. The revisions are incorporated as under:

p. 7, line 161 (Methods):

*"Here, we have sub-divided the available data records of each catchment into up to five individual 20-year periods **over the last century, denoted as $T_i$ (Table 1), where $T_i$ represents the $i^{th}$ 20-year period.** This 20-year period was chosen deliberately to balance the need for a sufficiently long period to minimize the impact of storage changes, while preserving the temporal sequence in the data that allowed us to place each catchment into a specific temporal stability category (as described in Step-4)."*

p. 7, line 170 (Methods):

*"Table 1: Symbols used in this study to present 20-year periods **($T_i$),** changes between subsequent 20-year periods, and distributions of deviations."*

*Reviewer comment:*

*L 503: "decrease in the seasonality": what does it mean*

Reply:

We thank the reviewer for highlighting the need for clarification regarding the decrease in seasonality. Seasonality refers to Seasonality Index (SI) of liquid precipitation input (rainfall + snowmelt), calculated by using the equation proposed by Gao et al. (2014), which describes how precipitation is distributed over the year. A higher SI indicates that most of the precipitation falls within a few months, while a lower SI value signifies a more even distribution throughout the year.

We have added a clarification of this point in the revised manuscript as follows:

p. 18, lines 497-500 (Results):

*"In contrast, 7 % of the catchments were tagged as "Alternating" and a dependency between $I_{E,i}$ and $\varepsilon_{IE\omega}$ could not be ruled out. A characteristic example for this type of catchments is the Kaituna*

catchment (New Zealand;  706 km$^2$, ID NZ_0000003) in Fig. 6j-l. This catchment features major fluctuations with median $\varepsilon_{IE\omega}$ between -0.115 and 0.198. In addition, although no systematic evolution of median $\varepsilon_{IE\omega}$ over time was evident (Fig. S4d), the data suggest the potential presence of a dependency on $I_{E,i}$ as shown in Fig. S4c. The pronounced alternating behaviour of the $\varepsilon_{IE\omega}$ fluctuations between -0.115 and 0.198, could not be readily explained by factors such as land use changes as estimated from the Hilda+ gridded land cover product (Winkler et al., 2021), **Seasonality Index (SI) of liquid precipitation input (i.e., rainfall + snowmelt), Parde Coefficients or median rainfall intensity (Fig. S3a,c,e-f). The SI was calculated using the formula proposed by Gao et al. (2014). A  higher SI value indicates that most of the precipitation falls within a few months, while a lower value reflects more evenly distributed precipitation throughout of the year.** This suggests that other additional drivers, or a combination of drivers, influence this catchment's alternating behaviour."

p. 18, lines 505-508 (Results):

"The remaining 102 catchments (4 %) were tagged as "Shift", as they exhibit a rather consistent shift of median $\varepsilon_{IE\omega}$ over time. This can be seen for a selected example in Fig. 6m-o.The median $\varepsilon_{IE\omega}$ in this catchment of the Zschopau River (Germany; 1544 km2; ID DE_0000027) does not only significantly vary between -0.055 and 0.037 but it does so rather systematically into one dominant direction after $\varepsilon_{IE\Delta1}$ ("+ - ++"; Fig. 6n). **This shift aligns with a gradual decrease in the 20-year Seasonality Index (SI) (Fig. S3e) of liquid precipitation input (i.e., rainfall + snowmelt). In the Zschopau river catchment, this decrease in SI towards the end of the century signifies a shift towards a more evenly distributed precipitation pattern. These changes coincide with an increase in forest cover towards the end of century, as estimated from Hilda+ data (Fig. S3b)**. Additionally, Renner et al. (2014) and Renner and Hauffe (2024) reported a gradual recovery of forests in the Zschopau catchment during this period, which may further contribute to the observed shift."